# Rigidly flat-foldable class of lockable origami-inspired metamaterials with topological stiff states

Amin Jamalimehr[1,3], Morad Mirzajanzadeh [1,3], Abdolhamid Akbarzadeh [1,2] & Damiano Pasini [1✉]

Origami crease patterns have inspired the design of reconfigurable materials that can transform their shape and properties through folding. Unfortunately, most designs cannot provide load-bearing capacity, and those that can, do so in certain directions but collapse along the direction of deployment, limiting their use as structural materials. Here, we merge notions of kirigami and origami to introduce a rigidly foldable class of cellular metamaterials that can flat-fold and lock into several states that are stiff across multiple directions, including the deployment direction. Our metamaterials rigidly fold with one degree of freedom and can reconfigure into several flat-foldable and spatially-lockable folding paths due to face contact. Locking under compression yields topology and symmetry changes that impart multi-directional stiffness. Additionally, folding paths and mixed-mode configurations can be activated in situ to modulate their properties. Their load-bearing capacity, flat-foldability, and reprogrammability can be harnessed for deployable structures, reconfigurable robots, and low-volume packaging.

[1] Department of Mechanical Engineering, McGill University, Montreal, QC, Canada. [2] Department of Bioresource Engineering, McGill University, Montreal, QC, Canada. [3] These authors contributed equally: Amin Jamalimehr, Morad Mirzajanzadeh. ✉email: damiano.pasini@mcgill.ca

O rigami and kirigami, the arts of folding and cutting paper, have inspired the development of a plethora of scale-invariant reconfigurable materials and structures that can deploy either spatially or in-plane[1,2]. These concepts have been implemented across disciplines, from mechanical memories[3], robotic actuators[4–8], thermally tunable structures[9,10], multistable devices[11–15], complex 3D geometries[16,17], and programmable surfaces[18–21] to flexible electronics[22,23]. Origami crease and kirigami cut patterns also proffer mechanical metamaterial designs with distinct geometric and mechanical properties, such as reconfigurability[24,25], flat-foldability[25–30], and bistable auxeticity[13,31] among others.

Of recent interest are in situ reprogrammable folding metamaterials[32–38] which harness an inherent coupling between the folding pattern and the geometry of motion. Here, rigid-foldability, flat-foldability, and load-bearing are distinct characteristics that can describe the modality of folding and the realization of certain functional performances. Rigid-foldability indicates that folding is solely controlled by the crease lines acting as rotational hinges, and not the deformation of the rigid panels[39]. Alternatively, in non-rigid-foldable patterns, both panel compliance and crease lines govern folding. Flat-foldability is a property that imparts a high level of reconfigurability by allowing spatial transformations leading into one or more flat states. Load-bearing in a foldable metamaterial simply denotes the capacity to offer structural resistance to a load applied in any given configuration across multiple directions.

Existing origami-inspired metamaterials offer a certain level of programmability, yet they are unable to attain concurrently rigid-foldability, flat-foldability, and load-bearing capacity along the deployment direction. One reason stems from the kinematics of their unit cell, which controls the way the crease pattern folds. Foldable metamaterials using the Miura-ori[27–29], interleaved[30], and tubular[13,26,40] patterns as well as cylindrical structures based on waterbomb patterns[35], and other unit cells, utilize crease geometry that exhibits some but not all of the properties defined above. For example, rigid-foldable material systems with multiple degrees of freedom (DoFs)[24,25], are either floppy or require precise control of the folding sequence, a characteristic that severely limits their capacity to withstand multidirectional loads. Alternatively, non-flat-foldable concepts[32,35] have limited reconfigurability, making their size and volume large, and most existing concepts utilizing structural instability[1,6,11,34,35] or the Kreseling pattern[33,41] to achieve reconfigurability are non-rigid-foldable. To fold, they must overcome a large energy barrier that bends and stretches their panels, thus sacrificing load-bearing capacity. On the other hand, foldable patterns that offer some load resistance can do so in certain directions only and mainly loses stiffness in the deployment direction[13,24–26,38,40,42]. This aspect can be problematic in applications where during service the load direction is uncertain, hence potentially reverting a stiff into a floppy configuration.

From the current state of the art, an interesting question arises: Can a crease pattern be conceived to reconcile the conflicting nature of rigid-foldability, flat-foldability, and load-bearing capacity in multiple directions, including that of deployment? This paper presents a framework for designing a topological class of rigidly flat-foldable metamaterials that are reprogrammable in situ to reconfigure along multiple directions, some flat-foldable and others lockable, where the latter is multi-directionally stiff even along the deployment direction. Our basis combines origami and kirigami concepts to introduce a crease pattern that is built cellular in its flat configuration, and subsequently stacked with the minimum number of layers to steer folding along one trajectory. To imbue reconfigurability, excisions are introduced in a crease pattern so as to relax the deformation constraints enacted by the rigidity of the faces of the parent origami, and to enable face contact within their intracellular spaces. Besides load-bearing capacity, our concept offers additional hallmarks including topology and symmetry switching that altogether enlarge the degree of in situ programmability. Finally, a simple yet effective fabrication process that can be easily automated is presented to impart three-dimensionality in the flat configuration.

## Results

**Geometry of reconfigurable unit chain.** To generate a rigidly-foldable unit chain that is flat-foldable and can lock into a stiff state upon face contact, we start from a primitive network of bars connected in a planar loop. The network is a planar $N$-bar linkage that forms a regular $N$ even-sided polygon. Figure 1a shows the generative process exemplified with a network of four bars, each of length $a$ and enclosing a square void. Extruding each bar outward to the length $b$ (red arrow) in the $x$-$y$ plane and at a given angle $\phi \in \left[\frac{\pi}{N}, \pi - \frac{\pi}{N}\right]$ yields four parallelograms, which we connect using isosceles triangles with a vertex angle $\lambda = \frac{2\pi}{N}$. By prescribing the folding profiles (dash lines) at each boundary between interfacing panels, we obtain a planar assembly of rigid surfaces that spatially fold along their connecting valley (V) and mountain (M) folding lines. The conceptual process can be thought as complementary to the fabrication steps (Supplementary Movie 1), where the void is first excised from a planar sheet of paper (kirigami-cuts), and then folded along prescribed dash lines (origami folds), thereby generating a hybrid architecture.

The fold lines of both V and M panels enable the system to act as a kinematic chain, in short, "unit chain". We define its configurational changes (Fig. 1a) using $m$ independent dihedral angles $\theta_1$, $\theta_2$, …, $\theta_m$. Each dihedral angle specifies the convex angle between the triangular panel and its adjacent quad panel (Fig. 1a), and $m$ denotes the mobility or nontrivial (term hereafter dropped) DoFs excluding rigid-body motions. We also assume the mountain and valley fold lines are constrained to remain on two parallel planes during folding. Explained later, this strategy is enforced through unit chain stacking, and it enables folding with a single DoF. In this case, the out-of-plane rise, $h$, is expressed as a function of three geometric parameters by:

$$h = a \sin\phi \sin\theta_1. \tag{1}$$

During folding, $\theta_1$ decreases and the unit chain can reach a lock state, denoted with the superscript "$L$", where contact between panels forbids any further motion (right sketch in Fig. 1a). In a lock state, the acute dihedral angles are given by:

$$\theta^L = \cos^{-1}\left(\cot\phi \tan\frac{\lambda}{2}\right). \tag{2}$$

The generative process illustrated in Fig. 1a for a unit chain with a square primitive void can be abstracted to other primitives, i.e., regular $N$ even-sided polygons, by merely varying $N$. This gives rise to a class of planar unit chains that spatially reconfigure within the voids and lock upon panel contact. Figure 1b illustrates five of them, where the first three rows show their initial fully developed state, the partially folded states, and one possible lock configuration. The last two rows depict their most compact in-plane tessellation and their out-of-plane stacking in paperboard specimens, where each layer is mirrored with respect to the $x$-$y$ plane and bonded at the triangular panels of the adjacent layers (above and below).

To investigate the kinematics of our unit chains, we first introduce the dimensionless extrusion factor $\bar{b} = b/a$ and adopt the notation $N_N(\bar{b}, \phi)$ to discriminate between unit chains. $N_N$ refers to the generic class of unit chains, where $\bar{b}$ and $\phi$ can assume any values. For the demonstrative purpose, we mainly

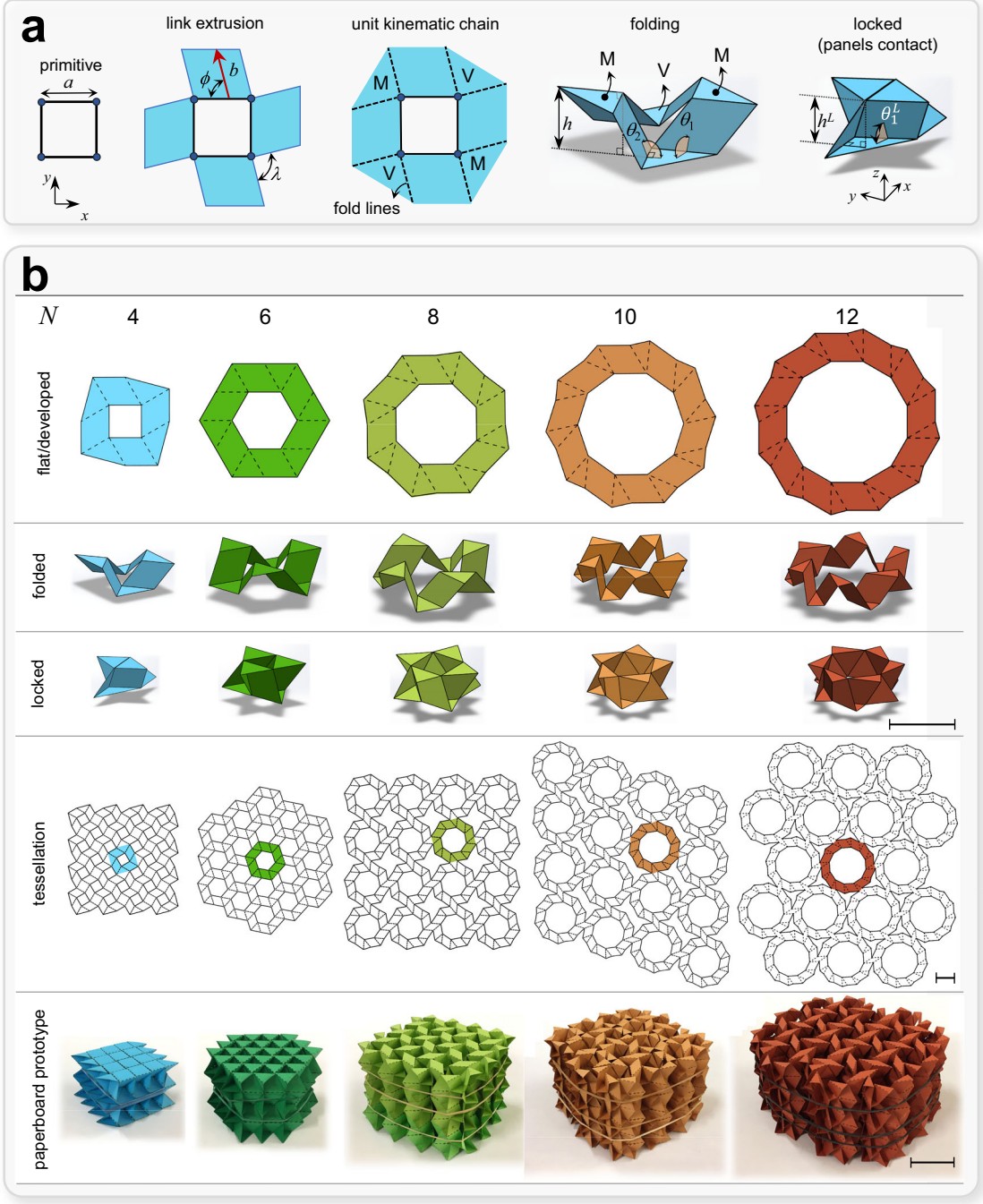

**Fig. 1 Conceptual scheme to generate self-locking kinematic chains and tessellations along with their paperboard prototypes. a** The building block of our reconfigurable class of rigidly flat-foldable materials. Primitive regular polygon describing a four-bar linkage, followed by in-plane angled ($\phi$) extrusion (red arrow of length $b$) of constitutive links; connection of extruded panels with isosceles triangular panels; addition of fold lines and assignment of a mountain (M) and valley (V). Upon folding, the unit chain reconfigures to reach its lock state (specified with "$L$") where contact is established between panels. **b** Concept abstraction to generate unit chains with an $N$ even-sided regular polygon for representative $N$, i.e., 4, 6, 8, 10, and 12. The first three rows show respectively the developed, partially folded, and one of the (multiple) lock states. The last two rows show the in-plane tessellation and their paperboard proof-of-concepts. The latter are multilayered spatial realizations obtained with the minimum number of layers, i.e., an even number of layers preserving their symmetry with respect to $x$-$y$ plane, that provides one degree of freedom. Rubber bands are here used to hold the unloaded configuration of the prototypes in their lock state, as the transition to the lock state requires the application of compressive forces in the $x$-$y$ plane, as explained later in the manuscript. Scale bars = 30 mm.

focus on a subset (Fig. 1b), namely the subclass defined by $\bar{b} = 1$ and $\phi = \pi/3$, i.e., $N_N(1, \pi/3)$ denoted hereafter as $\widehat{N}_N$ for simplicity, with $N = 4, 6, 8, 10$ and 12, and we refer to the more general case $N_N$ whenever our analysis become independent of the two geometric parameters $\bar{b}$ and $\phi$.

**Kinematic model**. We study the unit chain kinematics (Fig. 1a) through a set of assumptions. First, the panels are considered as infinitely rigid plates and the fold lines as rotational hinges. Second, to ease the formulation of the kinematic constraints, we replace the unit chain with a triangulated network of inextensible

elements connected through pin joints (Supplementary Fig. 1). Third, the edges of the triangular panels are modeled as bars, and quad panels are replaced with two triangles satisfying the planarity condition of their interplanar angles over the entire folding process (Supplementary Fig. 1d). By modeling our unit chain as a pin-jointed network of inextensible bars, we now examine its DoFs, and types of kinematic motion.

**Number of degrees of freedom**. The unit chain mobility is formulated through the rigidity matrix $\mathbf{R}$ pertinent to its structural assembly. The number of DoFs, $m$, for a pin-jointed triangulated network is given by $m = 3j - n_K - r$, where $j$ is the total number of joints, $n_K$ the number of external kinematic constraints, and $r$ the rank of $\mathbf{R}$ (see Supplementary Discussion "Kinematic analysis"). For our unit chain, $m$ also represents the number of independent dihedral angles. For the unit chain $\widehat{N}_4$ (Fig. 1a), $m = 5$, i.e., five mechanisms are possible. $m$ can be reduced to 1 if the mountain and valley fold lines are constrained to remain on two parallel planes during folding. The specific values and relationships the dihedral angles assume define distinct types of motion, as described below.

**Kinematic paths**. There are specific trajectories our unit chain can follow during motion. Defined as kinematic paths, each of them can be uniquely defined by a relation between the independent dihedral angles. For example, the equality of dihedral angles, i.e., $\theta_1 = \theta_2$, represents the type of motion shown in Fig. 1a, whereas $\theta_1 = \pi - \theta_2$ describes another one.

**Unit stacking as a pathway to reduce mobility**. If the triangular panels shown in Fig. 1a do not remain parallel during folding, our unit chain is endowed with manifold DoFs, which for $\widehat{N}_4$ is five. Having too many DoFs can be problematic. The unit chain can act as a multi-DoF mechanism with a tendency to be floppy such that folding cannot be easily controlled. One way to prune DoFs is to act along the third direction ($z$), and stack layers of unit chains one on top of the other. Figure 2a shows this strategy applied to $\widehat{N}_6$. The mountain facets of the triangular panels of the top unit are bonded to the valley facets of the adjacent unit (below). By stacking and bonding three unit chains, the mountain and valley triangular panels lie in parallel planes. If constructed from a material with sufficiently high elastic modulus, the non-negligible thickness of the triangular panels restrict the relative rotation range of two connected quad panels. This allows only equal and opposite rotations, which satisfy kinematic compatibility, and avoid encountering the energy barrier of the bending or stretching of the panels (Supplementary Discussion "Additional constraints" and Supplementary Fig. 2).

The strategy above is simple yet effective. Not only does it turn off DoFs, but also provides robust reconfiguration along a single folding path. Having now layers of unit chains, we can denote a generic multilayered $N$ even-sided unit chain with $n$ stacking layers by $N_N n_n$. Through a rigidity analysis of $N_N n_n$, we assess the role of layer stacking, $n$, on the DoFs (Supplementary Discussion "Kinematic analysis"). The outcome of this analysis is shown in Fig. 2b, where $\theta_i \neq \pi/2$ ($i = 1, 2, 3 \ldots m$) is assumed, and $\theta_i = \pi/2$ refers to the kinematic bifurcation. From the plot, we gather that the DoFs of single layer unit chains increase linearly with $N$ through the relation $m = 2N - 3$ which reduces to $m = 1$ if multiple layers are stacked, thus enabling reconfigurability along one single path. The minimum number of layers, $\bar{n}$, necessary to trim $m$ to 1 depends on the unit chain primitive. Figure 2b shows that for unit chains $N_4$ and $N_6$, $\bar{n} = 3$, whereas for $N_{N>6}$, $\bar{n} = 5$. With a focus on multilayered units with a single DoF, denoted by $N_N n_{\bar{n}}$, we now investigate their behavior at the instant of kinematic bifurcation and post-bifurcation.

**Kinematic bifurcation: emergence of lock and flat-fold modes**. In the early stage of folding, our $N_N n_{\bar{n}}$ chain can travel along one route governed by $\bar{n}$. However, as soon as it reaches a configuration where all its dihedral angles are $\pi/2$, i.e., kinematic bifurcation, the DoFs instantaneously grow, and multiple kinematic paths become active (Fig. 2a, c). We classify the post-bifurcation modes into either two-dimensional (2D) flat-foldable, or three-dimensional (3D) lockable (Fig. 2c). Flat-foldable modes are fully flat patterns that are distinct from the initial state. Lockable modes describe 3D states with contact between adjacent panels that impart compressive stiffness (Supplementary Movies 2 and 3).

We can show (Methods) that our single-DoF multilayered unit chains possess only two types of dihedral angle pairs (Fig. 2a, d), acute $\mathscr{A}$ and obtuse $\mathscr{O}$, obeying the relation $\mathscr{A} + \mathscr{O} = \pi$ during the entire range of motion. A given sequence with angle pairs, e.g., two $\mathscr{A}$s and one $\mathscr{O}$ depicted in Fig. 2d, can be simply denoted by the series of angle pairs, e.g., $\mathscr{A}\mathscr{A}\mathscr{O}$, and in compact form with the power indicating the repeated pairs, e.g., $\mathscr{A}^2\mathscr{O}$. This notation allows discriminating between kinematic modes that emerge at bifurcation. Given the modes defined by a repeating sequence of $\mathscr{A}\mathscr{O}$, such as $\mathscr{A}\mathscr{O}\mathscr{A}\mathscr{O}\mathscr{A}\mathscr{O}\ldots$ (or simply $\overline{\mathscr{A}\mathscr{O}}$), cannot be tessellated in-plane, we denote them as irregular modes as opposed to the regular ones which are tileable in-plane (Methods).

With the above, we systematically characterize the regular modes of a generic $N_N n_{\bar{n}}$ and determine the total number of possible reconfiguration modes. We use the Pólya enumeration theorem of combinatorics (Supplementary Discussion "Pólya enumeration theorem") to (i) count the regular modes and then (ii) define their kind. The problem of finding all independent regular modes of a generic $N_N n_{\bar{n}}$ unit can be treated as the classical necklace problem. The goal is to reconstruct the colored pattern of a necklace of several beads, each colored either in white or black ($\mathscr{A}$ or $\mathscr{O}$), from the knowledge of a limited set of information (Methods). By using a predictor-corrector type incremental method (Supplementary Discussion "Kinematic analysis"), we can now visualize the post-bifurcation kinematic paths of $\widehat{N}_4 n_4$, $\widehat{N}_6 n_4$, $\widehat{N}_8 n_5$ and $\widehat{N}_{10} n_5$ as shown respectively in Supplementary Movies 4–7.

To understand the relation between the total number of post-bifurcation modes and the sides of the unit chain primitive $N$, Fig. 2e plots the total number of lockable modes $c^L$ and flat-foldable modes $c^F$ versus $N$. The best curve fits are included to provide phenomenological closed-form relations that characterize folding paths and differentiate lockable modes ($c^L = 0.21 \exp(0.3N)$) from flat-foldable modes ($c^F = 0.54 \exp(0.16N)$) as a function of $N$ for a generic $N_N n_{\bar{n}}$. The results show that the number of regular modes grows exponentially, hence providing a rich platform to program lockable paths (Supplementary Table 1).

**Symmetry and topology**. From a single multilayered unit chain, we now turn to its periodic and multilayered tessellation forming a material system. Multilayered unit chains are connected to follow a tessellation pattern (Supplementary Discussion "Tessellations") that replicates a crystallographic arrangement of atoms. Figure 3 shows the top view of representative patterns for $N = 4$, 6, 8, and 10. While several others exist, here we focus on tessellations with the most compact pattern. The goal is to show that upon folding along a given path our material systems undergo switches in symmetry and apparent topology, both hallmarks of in situ programmability.

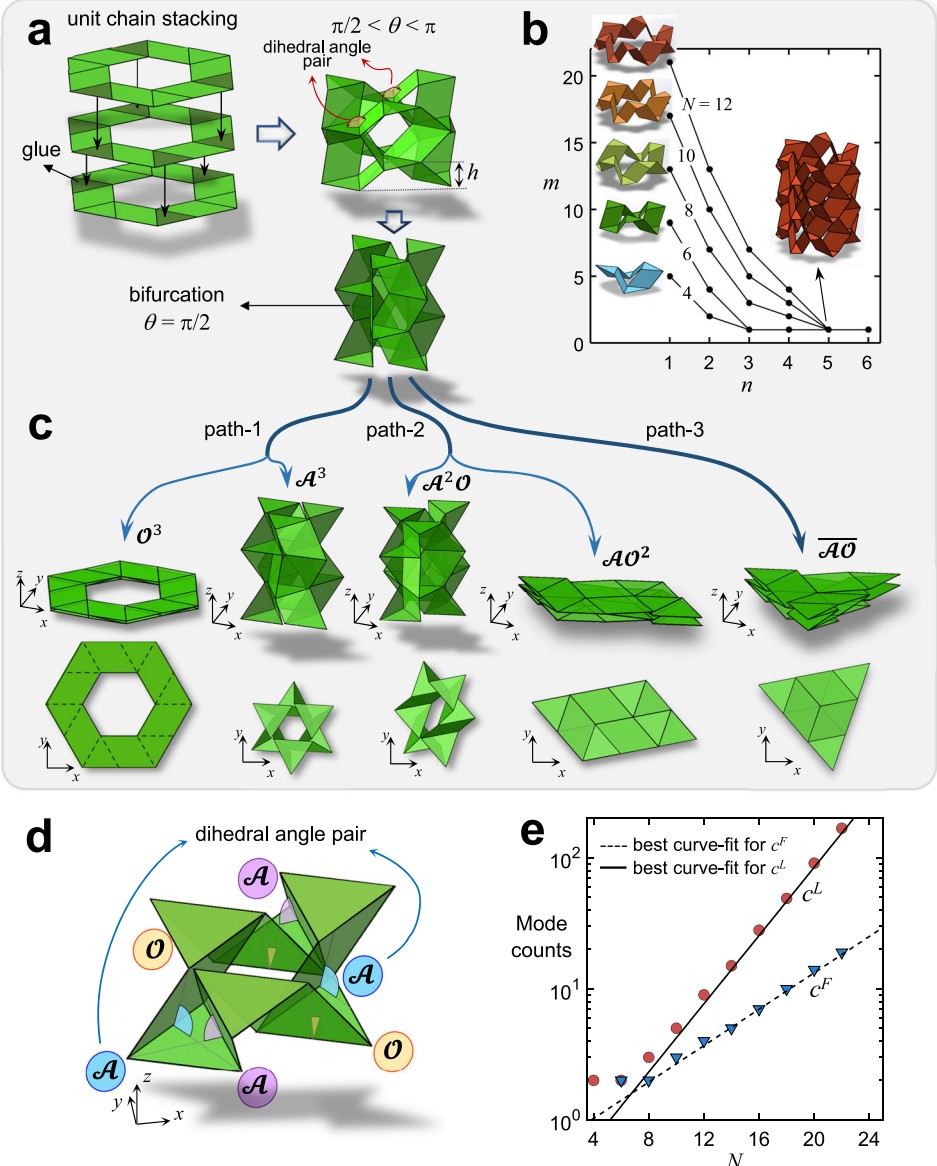

**Fig. 2 Multiple kinematic paths leading to lockable and flat-foldable modes. a** Unit chain stacking for a three-layered unit $\hat{N}_6 n_3$. On its right, a representative pre-bifurcation reconfiguration with $\theta < \pi$ (above) and the bifurcation state with $\theta = \pi/2$ (below). **b** Pre-bifurcation DoFs, $m$, plotted as a function of the number of stacked layers, $n$, for given $N$ even-sided primitives ($N = 4, 6, 8, 10, 12$). **c** Isometric and top views of five post-bifurcation modes belonging to three kinematic paths: four regular modes, of which two are locked ($\mathscr{A}^3$ and $\mathscr{A}^2\mathcal{O}$) and two flat-folded ($\mathcal{O}^3$ and $\mathscr{A}\mathcal{O}^2$), and one irregular mode ($\overline{\mathscr{A}\mathcal{O}}$). **d** Zoom on the top layer of $\hat{N}_6 n_3$ in lock mode $\mathscr{A}^2\mathcal{O}$ showing the arrangement of pairs of valley dihedral angles (in color): Three pairs of dihedral angles are illustrated here, two $\mathscr{A}$s (violet and blue) and one $\mathcal{O}$ (yellow) for a total of six dihedral angles. **e** Mode counts calculated from discrete data points for $N > 4$ along with the best curve fits characterizing the number of flat-foldable $c^F$ and lockable $c^L$ modes vs. $N$ for a generic multilayered unit $N_N n_{\bar{n}}$ (semi-log plot). (Denomination of the dihedral angles and modes: We specify acute angles with $\mathscr{A}$ and obtuse angles with $\mathcal{O}$. The geometry of the units enforces the condition $\mathscr{A}+\mathcal{O}=\pi$ during their entire range of motion. Since in $N_N n_{\bar{n}}$, which has one DoF, each pair contains equal angles, all $\mathscr{A}$s are equal as are all $\mathcal{O}$s. A given sequence with angle pairs, e.g., two $\mathscr{A}$s and one $\mathcal{O}$ depicted in **d**, can be simply denoted by the series of angle pairs, e.g., $\mathscr{A}\mathscr{A}\mathcal{O}$, and in compact form with the power indicating the repeated pairs, e.g., $\mathscr{A}^2\mathcal{O}$.)

Notions of crystallography become handy to study changes in symmetry. Each pattern is formed by tessellating in-plane a representative unit (red boundaries in Fig. 3) defined by the periodic vectors. Upon reconfiguration, the material system at bifurcation can access a new kinematic path that causes the smooth transition of certain dihedral angle pairs from $\mathscr{A}$ to $\mathcal{O}$; the result is a break in symmetry. This phenomenon is visualized by the patterns that followed those in the first column of Fig. 3, each denoted by their crystallographic point group and Schoenflies symbol. A variation in the lattice point groups translates into a change of the elastic constants defining the elastic tensor; the

symmetry shift endows the material system with another set of elastic properties. For example, the lattice made of $\hat{N}_4 n_{\bar{n}}$ (first row of Fig. 3) is shown in one of its lock modes, $\mathscr{A}\mathcal{O}$, in the second column. In this mode, $\mathscr{A}\mathcal{O}$ has $C_{2h}$ symmetry, i.e., a twofold rotational symmetry, and its elastic compliance matrix contains 13 elastic constants defining a monoclinic behavior. Upon switching to lock mode $\mathscr{A}^2$ (third column), its symmetry changes to $C_{4h}$, a fourfold rotational symmetry, resulting in an elastic compliance matrix with seven independent constants.

Besides symmetry, another aspect enabling in situ programmability is the apparent change of topology undergone after

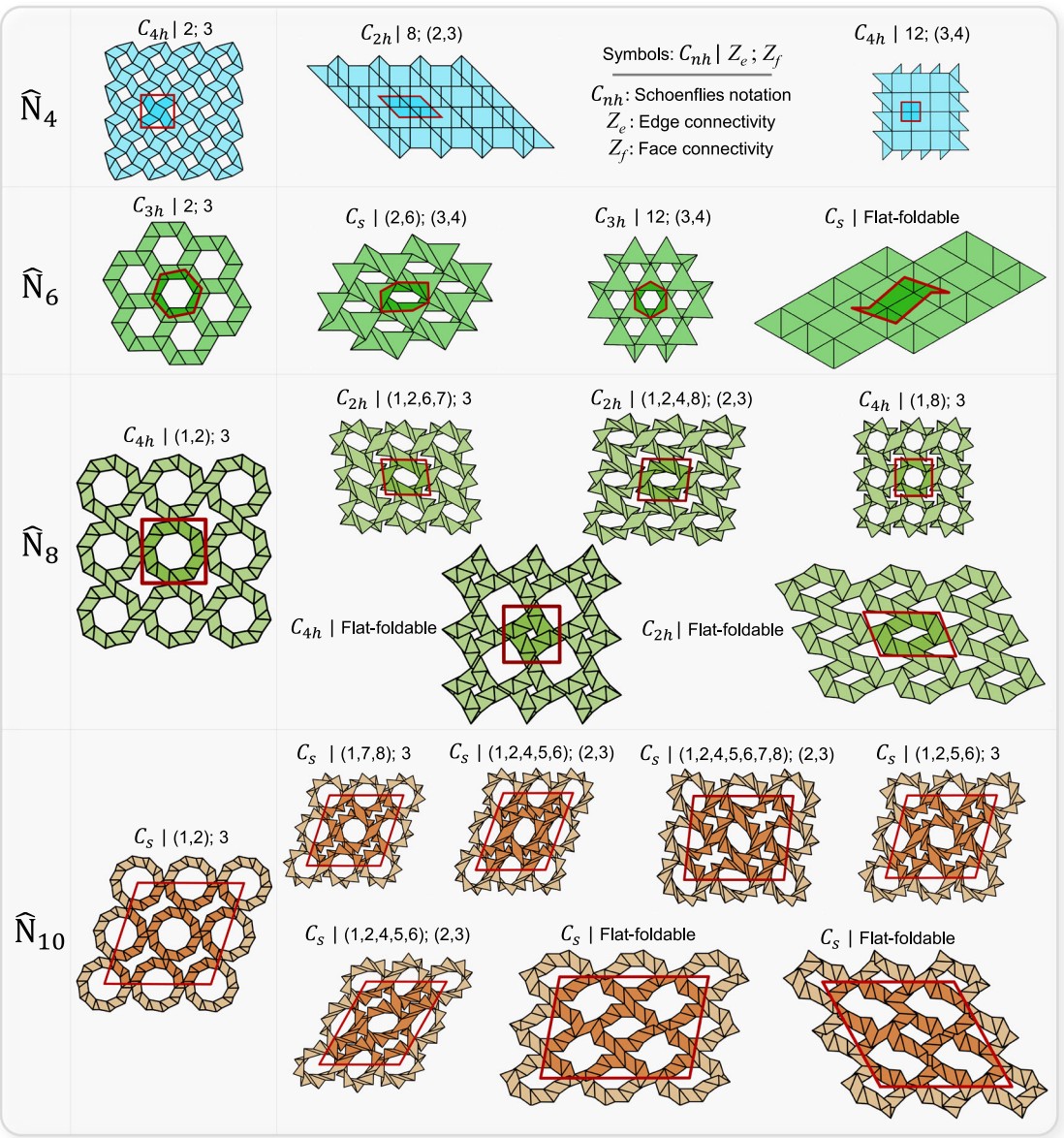

**Fig. 3 Topology and symmetry changes upon folding.** Regular flat-foldable and lockable modes of in-plane tessellated systems made of multilayered unit chains $\hat{N}_4 n_{\bar{n}}$, $\hat{N}_6 n_{\bar{n}}$, $\hat{N}_8 n_{\bar{n}}$, and $\hat{N}_{10} n_{\bar{n}}$ (qualitative sketches out-of-scale). Group points using Schoenflies notation are shown for all modes. $C_n$ refers to an $n$-fold rotation axis, $C_{nh}$ refers to $C_n$ with the addition of a mirror plane perpendicular to the axis of rotation, and $C_s$ denotes a group with only a mirror plane ($C_{1h}$). Edge and face connectivity are only shown for lockable modes. $Z_e$ is the number of edges that meet at a point (joint). $Z_f$ represents the number of faces that meet at an edge. The values of $Z_e$ and $Z_f$ for each configuration are shown after a vertical line separating them from their Schoenflies notation; parentheses are used whenever more than one connectivity value exists. For each configuration, multiple values of $Z_e$ and $Z_f$ may exist for a given joint and edge within the lattice. In these cases, all the values are reported in parentheses.

bifurcation. Here topology is defined by connectivity through $Z_f$, the number of faces that meet at an edge, which is equivalent to a fold-line as opposed to a cut-edge, and $Z_e$, the number of edges meeting at a vertex. The values of $Z_e$ and $Z_f$ are calculated for the corresponding spatial lattice upon the assumption that in the lock state coincident edges and vertices form a single edge and a single vertex (values on top of each configuration in Fig. 3). A change in $Z_e$ and $Z_f$ denotes topology differentiation, whereas topology is invariant to other geometric parameters, such as the length of the primitive void. Results show an increase in connectivity values from the partially folded state to the lock state (see also Supplementary Discussion "Rigidity"). A break in topology impacts the deformation mode of the panels, which transition from bending to stretching[43], as shown later. The versatility to impart topological

changes by traveling across a kinematic bifurcation can be used not only to tune the mechanical and physical properties in situ (see also Supplementary Discussion "Geometric mechanics of representative unit cell (RUC)", "Relative density" and "Poisson's ratio"), but also to switch the mechanism of deformation.

**Energy landscape.** Prior to bifurcation, our multilayered system $N_N n_{\bar{n}}$ folds with one DoF through the application of in-plane forces. At the point of kinematic bifurcation, however, multiple kinematic paths become accessible (Supplementary Movie 3). The entry into a specific path depends on the interplay between the components of the compressive forces, applied in its plane at the bifurcation instant. To study their role, we examine a generic

$N_N n_n$ (with $n \geq \bar{n}$) bi-axially and uniformly compressed and investigate the energy landscape of each kinematic path. The goal is to find the path and pertinent mode with the lowest energy level.

To formulate the total energy of $N_N n_n$ we assume infinitely rigid panels and frictionless rotational springs (hinges) of uniform stiffness per unit length, $k$. We consider our unloaded system in mode $\mathscr{O}^{N/2}$, with all dihedral angle pairs being obtuse; this is the zero-energy state of the system denoted by $\theta^0$ specifying a configuration that is either flat or partially folded due to the presence of residual stresses induced by manufacturing. $\theta^L$, on the other hand, is the acute dihedral angle of the lock unit. We assume $\theta^0 > \frac{\pi}{2}$, a condition implying that upon the application of an out-of-plane load ($z$-direction in Fig. 1a), our system can only fold from its zero-energy state to its fully developed configuration.

**Mode-phase diagram: the role of in-plane confinement.** We consider a pair of compressive in-plane forces, $f_x$ and $f_y$, applied uniformly, quasi-statically, and oriented along the principal directions $x$ and $y$ (Fig. 1a). We define the biaxiality ratio $r_B = f_y/f_x$ with $f_y \in [0, \infty)$, $f_x \in (0, \infty)$ and $r_B \in [0, 1]$ to discriminate between the relative magnitude of the applied forces, and derive an expression of the total energy as a function of the applied forces and the dihedral angle, i.e., $\Pi = \Pi(\theta, f_x, f_y)$ or $\Pi = \Pi(\theta, f_x, r_B)$ (Supplementary Discussion "Energy analysis").

Two representative systems $\widehat{N}_4 n_n$ and $\widehat{N}_6 n_n$ (with $n \geq \bar{n}$ and geometric parameter $\bar{b} = \frac{b}{a} = 1$) are examined. Their energy landscapes can be mapped into mode-phase diagrams (Methods) and their application showcased in demonstrative scenarios defined by families of applied in-plane forces. Each family is specified by $(f_x, r_B)$ where $r_B$ is maintained constant over the entire loading history. We consider two load families. The first is $(f_x, r_B = 1/3)$, where $f_x$ and $f_y$ can be proportionally scaled to respect the one third ratio; the (dimensionless) total energy landscape of one specific case ($f_x = 1, r_B = 1/3$) of that family is shown in Fig. 4a by the blue curve for a system with $n = 3$ and $\theta^0 = \pi$. In Fig. 4b, ($f_x, r_B = 1/3$) is shown by the blue load-path ABCD crossing three domains. In the light-yellow region, an increase of the force magnitudes from A to B is not sufficient to reconfigure our system, which remains flat ($\theta = \pi$) in its fully developed state. Once the load magnitude reaches the first domain boundary (black dash line), point B, the system starts to fold along the only kinematic path (yellow region) up to bifurcation (black solid line), point C. Immediately after bifurcation, the system can in principle access two modes ($\mathscr{AO}$ and $\mathscr{A}^2$ branches in Fig. 4a), yet it enters $\mathscr{AO}$ due to the lower energy it requires for activation. After entering in mode $\mathscr{AO}$, the system momentarily continues to reconfigure until it locks at point D (Fig. 4a, b). At this stage regardless of the magnitude of $f_x$ and $f_y$ satisfying $r_B = 1/3$, no further folding is possible as the panels have come into contact, thus imparting stiffness (Supplementary Movies 8, 9).

The second family of in-plane forces is ($f_x, r_B = 1$), shown in Fig. 4b by the green load-path AEFG. The total energy landscape of one specific load case ($f_x = 1, r_B = 1$) for that family is illustrated in Fig. 4a (green). Similar to ABCD, the system remains in its flat configuration for a load increase from A to E. A further increase of the in-plane confinement makes the system exit the initial region (black dash line) and gradually deform up to the bifurcation point F (second boundary threshold). Thereafter, it enters the lowest energy branch ($\mathscr{A}^2$ in Fig. 4a and light blue region in Fig. 4b) with an energetically stable state that occurs slightly prior to its lock state, point G. An additional load increase

makes the system fold until it reaches its lock state (dark blue region in Fig. 4b) (Supplementary Movie 10).

The map in Fig. 4b shows that the only way for $\widehat{N}_4 n_n$ to access mode $\mathscr{A}^2$ is with $r_B = 1$; in contrast, any other values of $r_B$ would bring the system to lock in mode $\mathscr{AO}$. This is an outcome that is distinctive to $\widehat{N}_4 n_n$, and does not necessarily translate to other systems. For instance, for $\widehat{N}_6 n_n$, more kinematic paths are available, i.e., multiple ranges of $r_B$ exist to switch between modes. Figure 4c shows the mode-phase diagram of $\widehat{N}_6 n_n$. Here, $r_B = 0.74$ (diagonal in the light green region) defines a condition for which the system, initially in $\mathscr{O}^3$ mode, enters $\mathscr{A}^3$ mode immediately after bifurcation. For $r_B > 0.74$, the unit enters its lockable mode $\mathscr{A}^2 \mathscr{O}$ (dark green), whereas for $r_B < 0.74$, it accesses its flat-foldable mode $\mathscr{O}^2 \mathscr{A}$ (brown region). Once entered in the post-bifurcation mode with minimum energy, the system remains in that region until it locks. At this stage, the system is rigid under compression and can no longer reconfigure; no other regions beyond the red boundaries (Fig. 4b, c) are accessible. The mode-phase diagrams depend on the geometric parameters $\bar{b}$ and $\phi$ only, the former being a nonlinear scale factor, and the latter governing the slope of the lines separating mode-regions and the loading path (biaxiality ratio $r_B$) of a given mode (Supplementary Discussion "Energy analysis"). They enable the choice of in-plane confining forces to attain desired post-bifurcation modes. The role of out-of-plane confinement is examined in the following.

**Lock state domains governed by out-of-plane force.** Our goal is to determine the magnitude of an out-of-plane ($z$ axis in Fig. 1a) compressive force $f_o$ necessary to bring and keep the system in its lock state without the need of in-plane confining forces.

For the demonstrative purpose, we examine $\widehat{N}_4 n_n$ folding in mode $\mathscr{A}^2$. Figure 4d schematically shows its energy curves (Eq. (3) in Methods) for three representing values of the out-of-plane load normalized by the lock load, i.e., $\bar{f} = f_o/f^L$ where the lock load $f^L$ is the minimum out-of-plane force at the lock state (Methods). The interplay between $f_o$ and $f^L$ gives rise to three domains:

Region I (light brown): $\bar{f} < 1$. Since $f_o < f^L$, the system cannot access the lock state from a given configuration.

Region IIa (yellow): $\bar{f} > 1$, $\frac{\partial \Pi}{\partial \theta} < 0$ and $\frac{\partial^2 \Pi}{\partial \theta^2} < 0$ – recall $\Pi$ is expressed as a function of ($\pi - \theta$). Partially folded at a given dihedral angle, the system is prone to fold back to its fully developed (flat) state.

Region IIb (blue): $\bar{f} > 1$, $\frac{\partial \Pi}{\partial \theta} > 0$, and $\frac{\partial^2 \Pi}{\partial \theta^2} < 0$. This is the lockable domain (Methods).

If in-plane forces lead the system to reach one unstable dihedral angle (blue point in Fig. 4d), a small perturbation prompts the system to naturally abandon it. Once the in-plane forces succeed in generating dihedral angles smaller than those described by Eq. (5) in Methods, our system can access the descending path in the lockable domain. Here spontaneous folding towards the lock state occurs, and in-plane forces are no longer needed. The magnitude of the out-of-plane action ($f_o > f^L$) enables lifting the in-plane confinement. Once in the lockable domain, e.g., orange point, the system is drawn to fold towards a stable configuration of equilibrium until it arrests due to panel contact (black point).

The analysis above has revealed the interplay between in-plane and out-of-plane confinement. The former can be imparted through the biaxiality ratio to program and steer the reconfiguration mode (either lockable or flat-foldable) during the folding process. The latter, in particular its magnitude ($f_o > f^L$) and the threshold value of the dihedral angle, i.e., the lockable domain boundary, set the conditions for spontaneous folding into the lock state without the need for in-plane compression. Once contact is

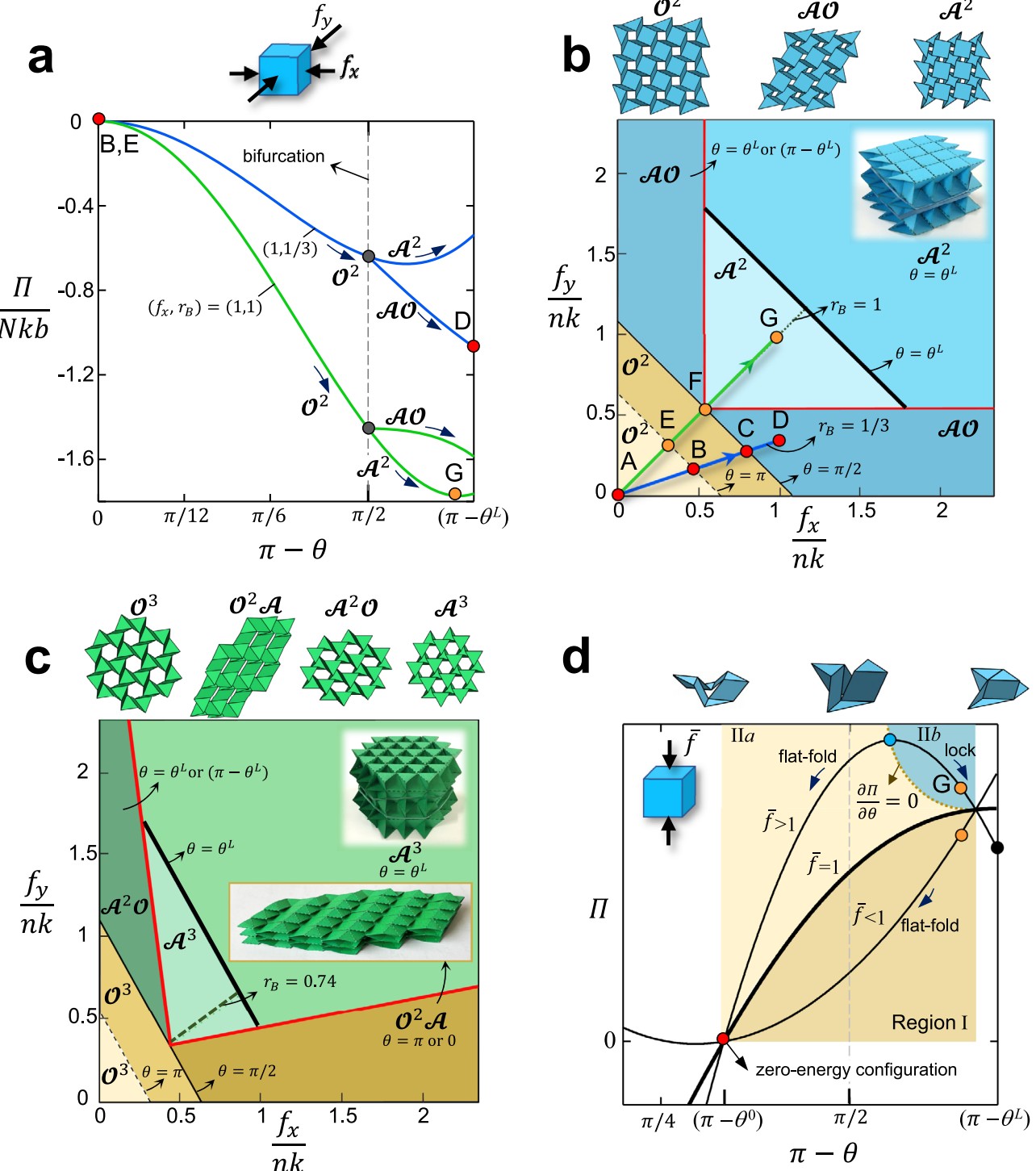

**Fig. 4 Energy landscapes and mode-phase diagrams. a** Dimensionless total energy landscape of $\hat{N}_4 n_{\bar{n}}$ unit subjected to two representative in-plane biaxial forces: $(f_x, r_B = f_y/f_x) = (1, 1/3)$, shown in blue, and $(1,1)$ shown in green. **b** Mode-phase diagram of $\hat{N}_4 n_{\bar{n}}$ unit subjected to dimensionless forces describing in-plane biaxial confinement $f_x/(nk)$ and $f_y/(nk)$, where $k$ is the rotational stiffness of the hinges per unit length: Family-load paths ABCD and AEFG describing uniform scaling of the applied forces. **c** Mode-phase diagram of $\hat{N}_6 n_{\bar{n}}$ unit under in-plane biaxial loads. Complementary information to plots **b** and **c** about the orientation of the lattice relative to the load direction is given in Supplementary Figs. 6 and 7. **d** Schematic total energy landscape of $\hat{N}_4 n_{\bar{n}}$ subjected to representative uniformly applied out-of-plane loads: $\bar{f}<1$, $\bar{f}=1$, $\bar{f}>1$, where $\bar{f} = f_o/f^L$. Three regions emerge for mode $\mathscr{A}^2$: Region I (light brown) shows the configuration space in which the unit under compressive load $\bar{f}<1$ folds in mode $\mathbb{O}^2$ towards a state in proximity to its zero-energy configuration; Region IIa (light yellow) shows the configuration space where the unit despite being subject to $\bar{f}>1$ still fold in mode $\mathbb{O}^2$ towards a state in proximity to its fully developed state; Region IIb (blue) illustrates the lockable domain, describing folding states where the unit has been brought by in-plane forces to a dihedral angle above the threshold of maximum energy (dot boundary); from this state the sole application of $\bar{f}>1$ makes the unit spontaneously folds in mode $\mathscr{A}^2$ until its panels contact.

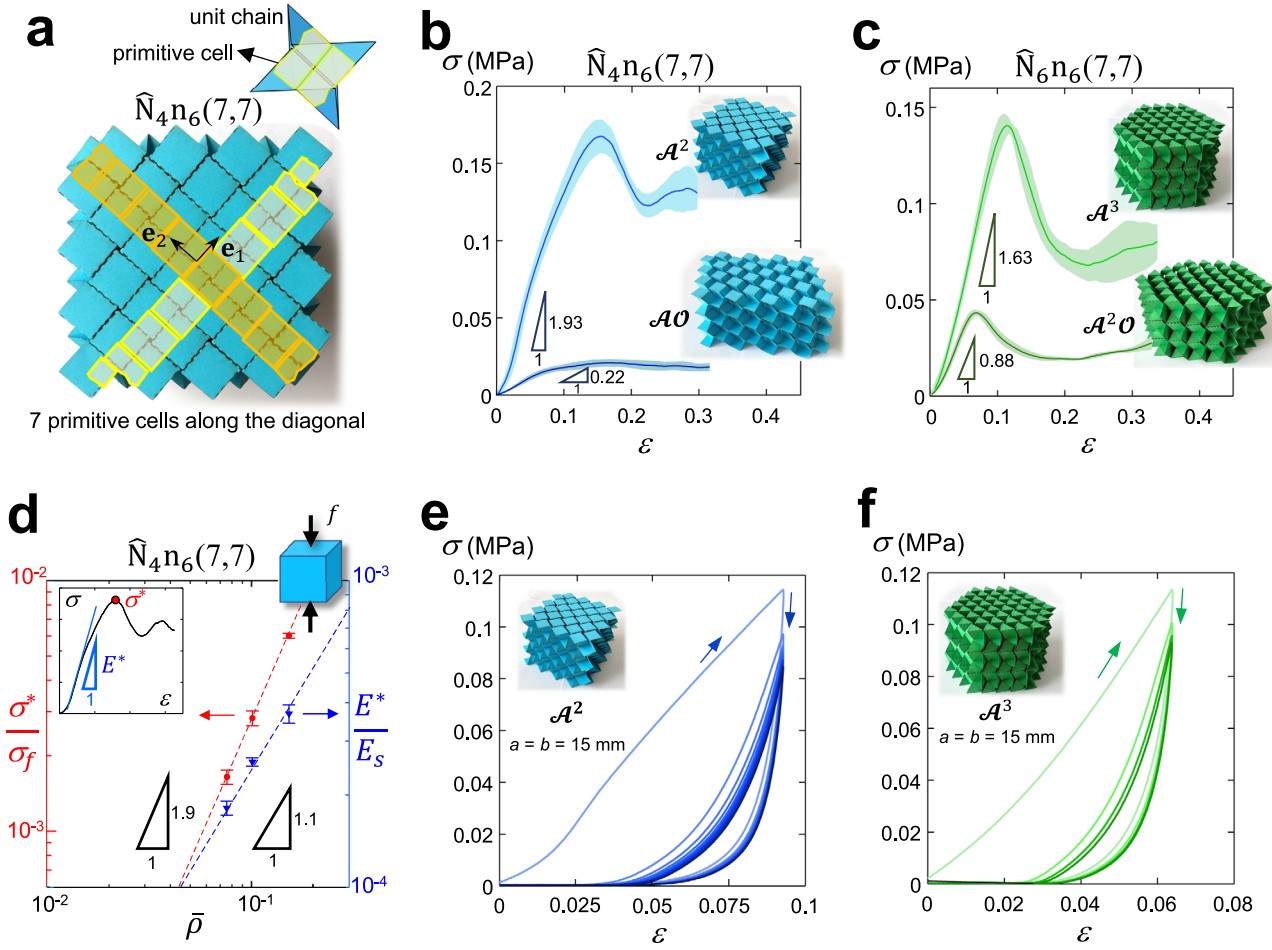

**Fig. 5 Out-of-plane compression response of cellular origami-inspired metamaterials.** Fabricated samples and measured properties of $\hat{N}_4 n_6(7,7)$ and $\hat{N}_6 n_6(7,7)$ unit in dissimilar lock modes; **a** Top view illustration of $\hat{N}_4 n_6(7,7)$ sample showing its primitive unit cell, and tessellation base vectors and tessellation levels. Engineering stress-strain curves measured for **b** $\hat{N}_4 n_6(7,7)$ in two lock configurations $\mathscr{A}^2$ and $\mathscr{A}\mathcal{O}$, and for **c** $\hat{N}_6 n_6(7,7)$ in two lock configurations $\mathscr{A}^3$ and $\mathscr{A}^2\mathcal{O}$. In **b** and **c** the shaded domain describes the dispersion of the results obtained from testing three samples. **d** Normalized compressive Young's modulus (triangle symbols) and normalized yield strength (circle symbols) of $\hat{N}_4 n_6(7,7)$ vs. relative density. Error bars represent the standard deviation of our measurements. The subset in **d** schematically illustrates how the compressive Young's modulus $E^*$ and yield strength $\sigma^*$ were measured. Cyclic (compressive loading-unloading) response of **e** $\hat{N}_4 n_6(7,7)$ (after ten cycles) and **f** $\hat{N}_6 n_6(7,7)$ (after four cycles) in their most compact lock modes $\mathscr{A}^2$ and $\mathscr{A}^3$, respectively, showing the cyclic response stabilizes nearly after four cycles. In these experiments, the unloading is performed at 75% of the compressive strength of the initial cycle of each sample.

attained, our system becomes a stiff structure (see Methods "Rigidity under compression" and Supplementary Discussion "Rigidity"), and it is ready to sustain compressive loads exerted in all three directions, as described below.

**Mechanical performance and programmability.** We examine a set of representative proof-of-concept specimens made of cellulose paperboard in their lock states under compression. The purpose is to demonstrate their capacity to achieve in situ distinct mechanical properties by uniform and nonuniform application of confining forces. Experimentally investigated are Young's modulus $E^*$ (the tangent of the compressive stress-strain curve in the first loading cycle assessing panel engagement as opposed to initial slippage) and the yield strength $\sigma^*$ (the first peak stress before densification); relative density and Poisson's ratio are given in the closed form in Supplementary Discussion "Relative density" and "Poisson's ratio". We avoid finite-size effects by tessellating in-plane $l_{\mathbf{e}_1} = l_{\mathbf{e}_2} = 7$ unit-cells along the base vectors $\{\mathbf{e}_1, \mathbf{e}_2\}$ and stacking $n = 6$ layers (see Fig. 5a and Supplementary Discussion "Experiments"). A generic tessellated system is thus denoted by $\hat{N}_N n_n(l_{\mathbf{e}_1}, l_{\mathbf{e}_2})$, and below two representative systems, $\hat{N}_4 n_6(7,7)$ and $\hat{N}_6 n_6(7,7)$, are examined.

*Load-bearing capacity.* Figure 5a shows the top view of $\hat{N}_4 n_6(7,7)$ along with an inset of a representative primitive unit cell (yellow) in the lock mode $\mathscr{A}^2$. Figure 5b reports its engineering stress-strain curves obtained in two lock configurations $\mathscr{A}^2$ and $\mathscr{A}\mathcal{O}$, and Fig. 5c the corresponding response of $\hat{N}_6 n_6(7,7)$ in its two lock configurations $\mathscr{A}^3$ and $\mathscr{A}^2\mathcal{O}$. The shaded domain describes the dispersion of the results obtained from testing three samples for each material system in a given mode. Three regions emerge: (i) An initial nonlinear regime, describing panels not yet engaged under compression, hence unable to establish a proper contact. (ii) A linear material response for both locked states of each system. (iii) The stress-peak and the following regime, both indicative of progressive panel buckling and creasing. For $\hat{N}_4 n_6(7,7)$ switching from $\mathscr{A}^2$ to $\mathscr{A}\mathcal{O}$, the Young's modulus and yield strength reduce approximately by an order of magnitude. For $\hat{N}_6 n_6(7,7)$ switching from $\mathscr{A}^3$ to $\mathscr{A}^2\mathcal{O}$, the corresponding

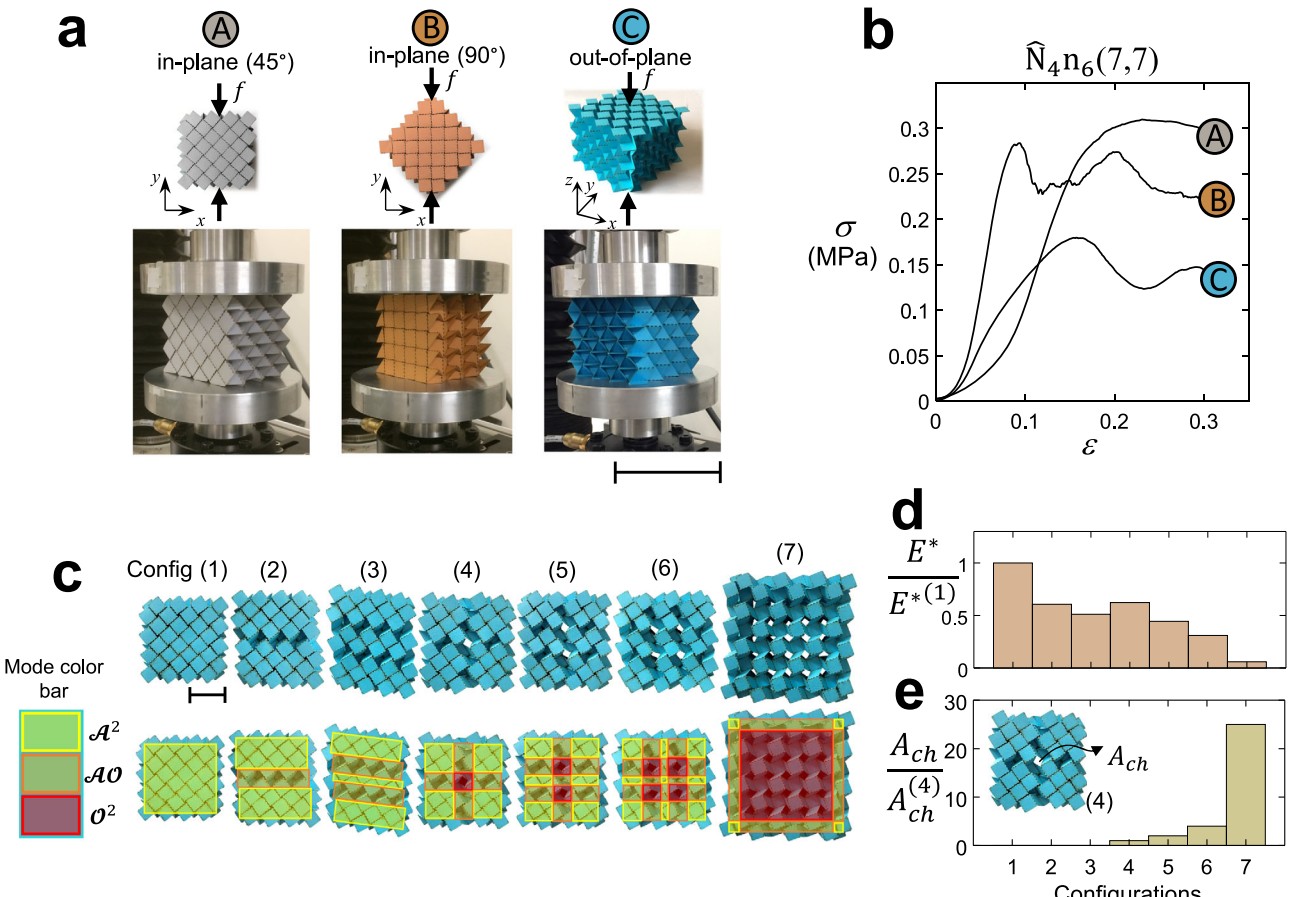

**Fig. 6 Multidirectional load-bearing capacity and in situ programmability. a** Directions (two in-plane and one out-of-plane) of the applied compressive loads relative to the orientation of the $\hat{N}_4n_6(7,7)$ specimen, and **b** their corresponding representative engineering stress-strain responses: Scale bar in **a** = 100 mm. **c** Top view of seven mixed-mode configurations of $\hat{N}_4n_6(7,7)$ with $a = b = 10$ mm: Scale bar in **c** = 30 mm. Regions of a given mixed-mode highlighted in semitransparent color for $\mathscr{A}^2$, $\mathscr{O}^2$ and $\mathscr{AO}$. Out-of-plane normalized compressive Young's modulus **d** and normalized open-channel (void) area in the out-of-plane direction **e** for the seven configurations shown in **c**.

reductions are approximately three and two times. Minor deviations emerge in regions (i) and (ii), as opposed to large values attained in region (iii) far from the peak. Our measurements are comparable with values reported for kirigami-based concepts made of similar material[44].

*Scaling laws.* Figure 5d shows the normalized Young's modulus, $\frac{E^*}{E_s}$, and normalized yield strength, $\frac{\sigma^*}{\sigma_f}$, (where $E_s$ and $\sigma_f$ are the Young's modulus and failure stress of the base material in the machine direction, MD) for $\hat{N}_4n_6(7,7)$ measured at three values of relative density. The results are obtained for samples with geometric parameters $a = b = 10$ mm, 15 mm, and 20 mm; for each of them, five identical samples were fabricated and tested. The normalized Young's modulus scales almost linearly with the relative density ($\frac{E^*}{E_s} \propto \bar{\rho}^{1.1}$), where $\bar{\rho} \propto \left(\frac{t}{a}\right)$ (Supplementary Discussion "Relative density"), hence obeying the classical scaling law of stretching dominated structures. The normalized yield strength, on the other hand, scales almost quadratically with relative density ($\frac{\sigma^*}{\sigma_f} \propto \bar{\rho}^{1.9}$), a scaling law deviating from the buckling or yield failure predictions of stretching dominated 3D plate-lattices. This can be attributed to the presence of additional failure mechanisms, which are governed by hinge stiffness, panel contact and the presence of geometrical imperfections, a topic of ongoing research.

*Cyclic response.* Figure 5e, f show the cyclic loading-unloading curves of $\hat{N}_4n_6(7,7)$ and $\hat{N}_6n_6(7,7)$ under compression. Given their similarity, only the response of $\hat{N}_4n_6(7,7)$ is examined. The hysteretic behavior might be attributed to the friction of faces and edges coming to contact, the viscoelastic-viscoplastic nature of the base material (cellulose paperboard) as well as the local accumulation of plastic damage, which is also responsible for progressive softening. This is caused by the repeated strain of the weak crease ligaments which amplify their detrimental effect at each cycle until the response stabilizes. This occurs after approximately four cycles. Thereafter, no appreciable softening can be observed at higher strains nor significant variations of the modulus. At the end of the test, a strain of 0.04 is registered, indicating a permanent set not fully recovered even after several hours from the test end. These characteristics are qualitatively comparable with those observed in soft polymeric lattices exhibiting viscoelasticity and localized plasticity under cyclic loading[45]. Further work is required to quantitatively assess the response we observed and the role of the governing factors.

*Multidirectional stiffness.* $\hat{N}_4n_6(7,7)$ in the lock configuration $\mathscr{A}^2$ is tested under in-plane and out-of-plane compression (Fig. 6a) along two in-plane directions (A and B) at 45° and 90° with respect to the $x$-axis, and along the $z$-direction. Representative curves of their engineering stress-strain responses in Fig. 6b attest a comparable yet distinct elastic response and load-bearing

capacity. The largest strength (A) and stiffness (B) observed during in-plane testing are attributed to the presence of double-layered panels, i.e., quad panels bonded to stack layers, an aspect that confers additional anisotropy and larger strength to bear the compressive load beyond the elastic regime. The difference between the initial in-plane responses (more compliant for the unit loaded at 45°) is due to the occurrence of a shear deformation that is dominant at the start of the compression test. Overall, our paperboard $\widehat{N}_4 n_6(7,7)$ specimens with $a = b = 15\,mm$ and locked in mode $\mathscr{A}^2$ weighs ~ 40 g, and can withstand up to 850 N under compression along the $z$-direction (Supplementary Movie 11), while up to 1450 N when compressed in the in-plane directions. In the lock mode $\mathscr{A}^3$, $\widehat{N}_6 n_6(7,7)$ with identical geometric parameters and weight can resist 1000 N under out-of-plane compression (Supplementary Movie 12).

*Mixed-mode configurations.* Uniformly applied forces at the instant of kinematic bifurcation cause the material system to fold into one of its lock modes, each with its own set of properties (e.g., Fig. 5b, c). Here we examine the outcome of a nonuniform set of forces locally applied in given zones, hence bringing the system into a mixed-mode configuration, i.e., a state that is partially folded and encompasses a combination of lockable and flat-foldable modes. Figure 6c depicts seven mixed-mode configurations among several others for our $\widehat{N}_4 n_6(7,7)$ specimen with square primitive side $a = b = 10\,mm$. The top row shows the emergence of voids from configurations 4 to 7. In the second row, the distribution of the attainable modes, i.e., $\mathscr{A}^2$, $\mathscr{O}^2$ and $\mathscr{A}\mathscr{O}$, which can concurrently form in a given mixed-mode, is highlighted with a given color. Figure 6d reports Young's modulus normalized with respect to that of the configuration (1), i.e., $E^{*(1)}$, measured in each mixed-mode configuration. Measurements of Young's modulus $E^*$ for $\widehat{N}_4 n_6(7,7)$ shows only a 1.2-fold decrease when the tessellation level (3,3) increases to (7,7) (see Supplementary Fig. 14b). The difference is small compared to the decrease observed for a switch from mode $\mathscr{A}^2$ to $\mathscr{A}\mathscr{O}$ (Fig. 5b) or $\mathscr{O}^2$ (Fig. 6d). This result suggests that the stiffness values of mixed-mode configurations are relatively insensitive to finite-size effects. In general, the addition of mode-regions $\mathscr{O}^2$ and $\mathscr{A}\mathscr{O}$ reduces stiffness, e.g., a drop to half is observed when configuration (1) switches to (3). Configuration (4) including all three mode-regions shows slightly higher stiffness than configuration (2), a counterintuitive result that might be attributed to the distribution of mode-regions with a twofold symmetry, as opposed to configurations (2) and (3), which exhibits reflection symmetry only. In this mode, only a single channel (an out-of-plane void) is formed as opposed to configurations (1), (2), and (3), which have no open channels or voids. Figure 6e shows the normalized open-channel area $\frac{A_{ch}}{A_{ch}^{(4)}}$ changes upon reconfiguration, where $A_{ch}$ is the open-channels area in the out-of-plane direction and $A_{ch}^{(4)}$ that of configuration (4) with a single channel. The change in the open-channel area can be considered as a descriptor of the system permeability, which scales linearly with the conduit area as in a porous medium. The trends show that the compressive Young's modulus and permeability are antagonists. While not quantified here, this qualitative result attests the versatility of our systems to tune on-the-fly flow permeability. Further work is required to quantify this aspect in detail.

## Discussion

This work has introduced a class of reprogrammable rigidly flat-foldable metamaterials that are topological and load-bearing in multiple directions including the deployment direction. By merging notions of origami-folding with kirigami-cuts, we have created foldable patterns made of chains that are shaped with an N even-sided regular polygonal primitives defining the inner void. Cellular excisions relax certain deformation constrains imposed by panel planarity and connectivity of the parent origami. The strategy enables folding within the embedded voids with multiple DoFs, which can be reduced upon stacking. This trait simplifies the fabrication process and eases its automation (Supplementary Discussion "Manufacturability" and Supplementary Movie 1). At the bifurcation instant, however, the DoFs grow, and multiple kinematic paths become temporarily accessible. Some lead to 3D stiff (locked) configurations, and others to 2D flat-folded states; this hallmark extends the level of programmability offered by existing origami concepts. When tessellated, our metamaterials undergo symmetry and topology transformations that are amenable to in situ modulation in uniform and mixed-mode configurations, each with its own set of properties.

Our foldable metamaterials offer functionalities that can be used as lightweight deployable and self-locking materials, low-volume transportable packaging, actuators, and lockable robotic systems that tune stiffness upon actuation. Their rigid-foldability also enables the adoption of stiff materials other than paper[39]. In addition, the load-bearing capacity in their densest lock configuration is not limited to any specific directions, rather it is offered along their three main directions, paving the way to their use as omnidirectionally structural, yet rigidly flat-foldable mechanical metamaterials. This aspect is distinct from current origami metamaterials featuring a trade-off between reconfigurability and load-bearing capacity, the latter being limited to certain directions[13,24–30,32,34,38,40,42]. These concepts when used to withstand volumetric pressure in their deployed state or another set of multidirectional forces, would collapse as they remain floppy at least in one direction. Volumetric pressure can be either externally or internally applied. Examples of the former include remotely operated vehicles, such as shape-changing vessels and submarines, made of structural components that need to deploy underwater and provide multidirectional stiffness and strength to resist water pressure during operation. Examples of the latter include inflatable systems, such as air tents, inflatable shelters, and buildings, saving boats and vests, which can not only be packed into a flat or other compact flat-folded configurations but also safely maintain their deployed state if punctured. Our metamaterials can thus work as the skeleton of puncture-resistant inflatable systems that lock in place after deployment, without collapse or losing their functionality due to unforeseen deflation. Furthermore, the capacity to switch permeability while remaining stiff can find application in the design of adaptive porous media and breathable walls for civil engineering, or as smart-valves for medical implants where fluid flow could be modulated in situ by structure rather than by external occlusions.

## Methods

**Identification of lockable and flat-foldable modes.** We systematically study the relations that define the post-bifurcation modes belonging to a given kinematic path for a single-DoF stacked unit chain $N_N n_{\bar{n}}$. As illustrated in Fig. 2a, in the pre- and post-bifurcation stages, the triangular panels (six in dark green in the middle layer) in each set of mountains and valleys remain parallel as $\bar{n} = 3$. Three of them are mountain panels that lay in a mountain plane, and the others rest in a valley plane. The distance between them is $h$ (Fig. 2a) which can be calculated through relation (1). In a given plane, there are six fold lines (two per triangle) which form a total of six dihedral angles. The fold lines that are parallel form a pair of dihedral angles (Fig. 2a). This pair can contain either equal or supplementary angles, a condition that defines the type of post-bifurcation mode. Equality of dihedral angles in each pair defines regular modes, which in Fig. 2c belong to paths 1 and 2; this implies that the reconfiguration of the unit chain leads to a folded pattern that is compatible with its original tessellation. In contrast, supplementarity of dihedral angles in all pairs, i.e., the dihedral angles sum up to 180° in each pair, gives rises to irregular modes, and path 3 in Fig. 2c shows an example. Irregular modes can be attained only in a single unit chain but not in a tessellated pattern, as they forego

folding congruence between the initial and the final pattern, revealing that the flat-foldable tessellation cannot be unpacked to its initial pattern. Only one irregular mode exists for $N_N n_{\bar{n}}$ with $N > 4$ and $N/2$ equal to an odd number, e.g., modes of $\widehat{N}_6 n_3$ shown in Fig. 2b.

Given the folding incompatibility of irregular modes, we now focus on the regular counterparts and study the conditions that can be used for a given kinematic path to count the number of existing modes and define the characteristics of each of them. Our goal is to demonstrate the existence of relations between distinct pairs of dihedral angles, which in turn govern the kinematic paths $N_N n_{\bar{n}}$ can access post-bifurcation. We first introduce some basic notions for our analysis. In the denomination of dihedral angles, we specify acute angles with $\mathscr{A}$ and obtuse angles with $\mathscr{O}$. The geometry of the units enforces the condition $\mathscr{A} + \mathscr{O} = \pi$ during their entire range of motion. As an example, three pairs of valley dihedral angles are illustrated in Fig. 2d, two $\mathscr{A}$s (violet and blue) and one $\mathscr{O}$ (yellow) for a total of six valley dihedral angles. Since in $N_N n_{\bar{n}}$, which has 1 DoF, each pair contains equal angles, all $\mathscr{A}$s are equal as are all $\mathscr{O}$s. A given sequence with angle pairs, e.g., two $\mathscr{A}$s and one $\mathscr{O}$ depicted in Fig. 2d, can be simply denoted by the series of angle pairs, $\mathscr{A}\mathscr{A}\mathscr{O}$, and in compact form with the power indicating the repeated pairs, e.g., $\mathscr{A}^2\mathscr{O}$. This notation allows discriminating between kinematic modes that emerge at bifurcation. For example, $\widehat{N}_6 n_3$ (Fig. 2c) can travel along four regular modes: $\mathscr{O}^3$ (three obtuse angles are engaged only), $\mathscr{A}^3$ (three acute angles are engaged only), $\mathscr{A}^2\mathscr{O}$ (two acute angles and one obtuse angle), and $\mathscr{A}\mathscr{O}^2$ (one acute and two obtuse), and one irregular, shown with $\overline{\mathscr{A}\mathscr{O}}$. Modes containing identical pairs of dihedral angles belong to the same kinematic path, and we designate them by swapping $\mathscr{A}$s and $\mathscr{O}$s, i.e., $\mathscr{A}^2\mathscr{O}$ and $\mathscr{A}\mathscr{O}^2$ belong to path 2, and $\mathscr{A}^3$ and $\mathscr{O}^3$ to path 1.

With the notions above, we can now systematically characterize the regular modes of a generic $N_N n_{\bar{n}}$ and determine the total number of possible reconfiguration modes. The problem of finding all independent regular modes of a generic $N_N n_{\bar{n}}$ unit can now be treated as the classical necklace problem. In our case, the equivalent necklace is our unit chain $N_N n_{\bar{n}}$ with $N/2$ colored beads (Fig. 2d), and each color represents a type of dihedral angles, either $\mathscr{A}$ or $\mathscr{O}$. By applying the Pólya enumeration theory, we determine all reconfiguration modes our $N_N n_{\bar{n}}$ chain can attain from knowledge of $N/2$ numbers of $\mathscr{A}$ and $\mathscr{O}$. We also assume that the beads can be rotated around the necklace and that the necklace can be flipped over. By applying this theory to $\widehat{N}_6 n_3$, for instance, all the possible modes can be collected in a generating function of the form $\mathscr{A}^3 + \mathscr{A}^2\mathscr{O} + \mathscr{A}\mathscr{O}^2 + \mathscr{O}^3$, which describes the four regular modes illustrated in Fig. 2, and where the sum of the powers of $\mathscr{A}$ and $\mathscr{O}$ in each mode is $N/2$. Similar results can be obtained for other unit chains, $N_N n_{\bar{n}}$ (see Supplementary Discussion "Pólya enumeration theorem" and Supplementary Fig. 3 for more details).

**Energy of the in-plane confinement and mode-phase diagram**. We start with $\widehat{N}_4 n_n$ described by the representative set of parameters: $k = 1/3$ N, $n = 3$, $\theta^0 = \pi$ and $a = b = 15$ mm. Figure 4a illustrates two typical curves of the dimensionless total energy, each for a given value of $(f_x, r_B)$. Upon bifurcation, when $\theta = \pi/2$, the total energy curve splits into two branches, each representing a reconfiguration mode $\mathscr{A}\mathscr{O}$ and $\mathscr{A}^2$. Between these two, the system chooses the mode which has the lowest energy level. This outcome can be determined by examining (i) the magnitude of the energy of all branches immediately before and after the bifurcation, and (ii) the gradient of the energy of all branches at bifurcation, for example, $\left.\frac{\partial \Pi_{\mathscr{A}^2}}{\partial \theta}\right|_{\theta = \pi/2}$ and $\left.\frac{\partial \Pi_{\mathscr{A}\mathscr{O}}}{\partial \theta}\right|_{\theta = \pi/2}$ (see Supplementary Discussion "Energy analysis" for more details). The magnitude and the ratio of the in-plane biaxial loads, i.e., $(f_x, r_B)$, govern the relative energy level of each energy branch, dictating the configuration mode our system would travel after bifurcation. For example, Fig. 4a shows the role of $r_B$ in entering a given post-bifurcation mode. For a load case $(f_x, r_B) = (1, 1/3)$, the system at bifurcation chooses the $\mathscr{A}\mathscr{O}$ energy branch until reaching the lock state in this mode; in contrast once subjected to $(f_x, r_B) = (1, 1)$, the system follows the $\mathscr{A}^2$ energy branch to reach the lock state.

The example above suggests the prospect to generate a mode-phase diagram that maps the activation of a given mode with respect to the relative magnitude of the in-plane confinement forces $f_x$ and $f_y$. Figure 4b visualizes such a map for $\widehat{N}_4 n_n$ with $a = b = 15$ mm and $\theta^0 = \pi$. Each color is assigned to a region that describes a given configuration mode. The boundaries separating modes $\mathscr{A}\mathscr{O}$ and $\mathscr{A}^2$ are obtained by equating the gradient of the total energy of each branch at bifurcation, $\left.\left(\frac{\partial \Pi_{\mathscr{A}^2}}{\partial \theta} = \frac{\partial \Pi_{\mathscr{A}\mathscr{O}}}{\partial \theta}\right)\right|_{\theta = \pi/2}$.

**Formulation of the energy of the out-of-plane confinement**. We start by expressing the potential energy of the hinges as $V = V(\theta) = 2nNkb(\theta - \theta^0)^2$ and the work of a uniformly applied external force $f_o$ as $W_o = nf_o a \sin\phi(\sin\theta^0 - \sin\theta)$, where we recall $n$ is the number of stacked layers and the other parameters are defined in Fig. 1a. The potential energy due to gravity is neglected since the panels of our system are made of cellulose paperboard, lightweight material with the gravitational potential energy of few orders of magnitude lower than that of the hinges and the work of the external forces. If we introduce $V' = V(\pi - \theta)$, and denote a generic mode with $\mathscr{A}^{\xi}\mathscr{O}^{\xi'}$, where $\xi$ is the total number of valley pairs for

the acute dihedral angles ($\mathscr{A}$) and $\xi'$ counts the total number of valley pairs for the obtuse dihedral angles ($\mathscr{O}$) with $\xi + \xi' = \frac{N}{2}$, we can express the total energy $\Pi = \Pi(\theta, f_o)$ of the $\mathscr{A}^{\xi}\mathscr{O}^{\xi'}$ mode as

$$\begin{cases} \Pi = \frac{2\xi}{N}(V - V') + V' - W_o & \forall \xi < 2 \quad \frac{\pi}{2} \le \theta \le \pi \\ \Pi = \frac{2\xi'}{N}(V' - V) + V - W_o & \forall \xi \ge 2 \quad \theta^L \le \theta \le \frac{\pi}{2} \end{cases} \tag{3}$$

The first expression in Eq. (3) describes the total energy of the system in mode $\mathscr{A}^{\xi}\mathscr{O}^{\xi'}$ prior to bifurcation, while the second gives the total energy after bifurcation.

Under an out-of-plane force, a generic system $N_N n_n$ is in equilibrium at the lock state when the total energy has a stationary value. Given $N_N n_n$ has one DoF, and the total energy and its derivative are continuous functions, we can determine the minimum out-of-plane force $f^L$ at the lock state by solving $\left.\frac{\partial \Pi(\theta, f_o)}{\partial \theta}\right|_{\theta = \theta^L} = 0$ for $\xi \ge 2$, which yields:

$$f^L = -4kb \frac{2\xi(\pi - 2\theta^0) - N(\pi - \theta^L - \theta^0)}{a \sin\phi \cos\theta^L}. \tag{4}$$

Equations (3) and (4) can be used to map the total energy landscape of a system under a uniform out-of-plane compression as a function of the supplementary of the dihedral angle $\theta$, i.e., $\pi - \theta$ (Fig. 1a). For demonstrative purpose, we examine $\widehat{N}_4 n_n$ folding in mode $\mathscr{A}^2$ with $\theta^0 = 2\pi/3$. Figure 4d shows its energy curves (Eq. (3)) for three representing values of the out-of-plane load normalized by the lock load, i.e., $\bar{f} = f_o/f^L$. Setting $\bar{f} = 1$ yields the boundary (thick curve) between two energy domains, one (below) satisfying $\bar{f} < 1$ and the other (above) $\bar{f} > 1$. The red point, which all curves (three shown) pass through, represents the zero-energy state, described above as the state of the system immediately after manufacturing, either flat (ideal case) or marginally folded due to residual stress from fabrication.

Subject to uniform out-of-plane compression, our material system can fold into its lock state under two conditions. First, the magnitude of the uniformly applied force $f_o$ should be above the minimum out-of-plane force, $f^L$, required to lock up the unit. Second, the dihedral angle of our unit should be larger than a threshold value defined by the maximum energy barrier of the system. The interplay between $f_o$ and $f^L$ described by these conditions gives rise to three domains:

Region I (light brown): $\bar{f} < 1$. Here fall configurations are defined by supplementary angles for which our system can reach an equilibrium that is either stable if $\frac{\partial^2 \Pi}{\partial \theta^2} > 0$, or unstable if $\frac{\partial^2 \Pi}{\partial \theta^2} < 0$. Since $f_o < f^L$, the system cannot access the lock state from a given configuration, e.g., lower orange point, and it tends to fold back to its equilibrium point along the "flat-fold" direction towards the zero-energy point (red).

Region II: $\bar{f} > 1$. In this domain, our system can potentially reach the lock state, but a difference in the outcome exists as determined by the stability of equilibrium. Region II splits into two subdomains (IIa and IIb), each defined by the slope of the energy curve, i.e., the sign of $\frac{\partial \Pi}{\partial \theta}$, where—we recall—$\Pi$ is expressed as a function of $(\pi - \theta)$.

Region IIa (yellow): $\frac{\partial \Pi}{\partial \theta} < 0$ and $\frac{\partial^2 \Pi}{\partial \theta^2} < 0$. The condition of equilibrium here is unstable despite the out-of-plane force being larger than the minimum locking force. In this region, a system partially folded at a given dihedral angle by the applied in-plane forces is prone to fold back to its fully developed (flat) state.

Region IIb (blue): $\frac{\partial \Pi}{\partial \theta} > 0$ and $\frac{\partial^2 \Pi}{\partial \theta^2} < 0$. This is the lockable domain, bounded by the locus of points (dot line), which satisfies the condition $\frac{\partial \Pi}{\partial \theta} = 0$ for all $\bar{f} > 1$. Upon imposing the condition $\frac{\partial \Pi(\theta, f_o)}{\partial \theta} = 0$ in Eq. (3), we can express the dihedral angle $\theta$ as a function of the load $f_o$ when the energy is maximum, $\frac{\partial \Pi}{\partial \theta} = 0$, from which we obtain

$$f_o(\theta) = -16kb \frac{(\theta - \theta^0)}{a \sin\phi \cos\theta}. \tag{5}$$

Substituting Eq. (5) into Eq. (3) yields the lockable domain boundary of maximum energy (dot bound in Fig. 4d) given as a function of the dihedral angle ($\theta^L \le \theta \le \frac{\pi}{2}$ and $\xi = 2$):

$$\Pi = V(\theta) + 16nkb \frac{(\theta - \theta^0)}{\cos\theta}(\sin\theta^0 - \sin\theta). \tag{6}$$

Equation (6) traces points of the dot boundary that are unstable configurations of equilibrium, where the total energy attains maximum values, one of which is shown by the blue point of the representative energy curve $\bar{f} > 1$.

**Rigidity under compression**. Once folded into the lock state, our multilayered unit chain $N_N n_{\bar{n}}$ inherits compressive load-bearing capacity as panels reach contact and prevent further motion. We study the load-bearing capacity of our class of foldable material systems in relation to their layer stacking. The condition that guarantees their structural rigidity in one of their lock states can be determined by studying their pin-jointed counterpart made of a triangulated network (Supplementary Fig. 1c). We can formulate the general problem that predicts the rigidity of a structure by theoretical analysis (Supplementary Discussion "Kinematic analysis" and the "Kinematic model" section). While the units are subjected to compressive loads, we assume coincident bars as a single bar and multiple coincident joints as a

single joint. The results can be expressed for a single unit chain as a function of the number of bars and joints at its lock and partially folded configurations along with the conditions of rigidity (see Supplementary Table 3 in Supplementary Discussion "Rigidity").

Carried out for the general lock state $\mathscr{A}^{N/2}$, where all dihedral angles are acute, our rigidity analysis reveals that $N_4$ becomes rigid with a single layer, whereas at least two layers must be stacked for $N_6$ and three layers for $N_{N>6}$. Knowing the minimum number of stacked layers provides an essential guideline to make our unit chain stiff in a lock state under compression.

**Experimental methods.** Test samples were built out of cellulose paperboard material (200 g m$^{-2}$ Fabriano Craft paper) with a dry thickness $t = 0.21$ mm. Each flat sheet was perforated via laser cutting (CM1290 laser cutter, SignCut Inc.) along prescribed patterns, followed by manual-folding and layer bonding (using a commercial Polyvinyl acetate, commonly known as white glue). Fold lines were obtained with cuts of 2 mm length spaced uniformly at 1.4 mm intervals. Experiments were performed with a BOSE ElectroForce-3510 tester (Bose Corporation, Framingham, Massachusetts) and a BOSE load-cell with a load capacity of 12.5 kN. Displacement-control was used with a quasi-static ramp loading and strain rate of $10^{-3}$ s$^{-1}$. Manufacturing and testing were performed at room temperature (22 °C) with a relative humidity of about 30%. To measure the properties, e.g., Young's modulus, we used both the displacement data obtained from the crosshead displacement readings as well as Digital Image Correlation (CCD camera—Point-Grey) for comparative purposes. The largest difference in measurements provided by the two techniques was below 3%. Young's modulus and yield strength were measured as the initial (linear region) slope (maximum strain at 0.05) and the first peak in the stress-strain curve, respectively, for the first loading cycle (see subset in Fig. 5d). Cyclic loading-unloading experiments were carried out using a displacement-control module at the same strain rate. The direction of the load was reversed when the load reached ~75% of the first cycle peak load.

All samples were realized with the following geometric parameters $a = b = 15$ mm and $\phi = 60°$ and loaded in the out-of-plane in compression unless stated otherwise. The relative density is defined by $\bar{\rho} = \rho^*/\rho_s$ (density of our reconfigurable cellular material, $\rho^*$, divided by the density of the solid material, $\rho_s$). To vary relative density, three values were used $a = b = 10$, 15, and 20 mm. Specimens were compressed between two smooth flat platens of Aluminum. Initially, samples were brought to their lock position manually and kept in this configuration by wrapping them up with a rubber band. To assess the role of the rubber band during the experiments, we repeated a few compression tests without a rubber band. In these instances, we started the experiment while the sample was wrapped with a rubber band, and then cut it after applying 2% of the peak load, i.e., ~28 N for $\hat{N}_4n_6(7,7)$. The results attest that the stress-strain curves were almost identical to those obtained on samples confined with a rubber band for the entire duration of the tests. See also Supplementary Discussion "Experiments" for further details about the properties of the paperboard.

## Data availability

The data that support the findings of this study are available from the corresponding author upon reasonable request.

## Code availability

The codes that support the findings of this study are available from the corresponding author upon reasonable request.

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

## Acknowledgements

D.P. and A.A. acknowledge financial support from the Natural Sciences and Engineering Research Council of Canada and Canada Research Chairs Program, and M.M. and A.J. from McGill University (McGill Engineering Doctoral Award) and Fonds de recherche du Québec—Nature et technologies.

## Author contributions

A.J., M.M., and D.P. designed research; A.J. proposed unit cells N4 and N6 and M.M. generalized them to the whole class; A.J. and M.M. performed research under close supervision of D.P.; M.M. and D.P. wrote the paper; all authors contributed to the interpretation of the results and the editing of the paper.

## Competing interests

The authors declare no competing interests.
