## [Peer Review File · Nature Communications]

REVIEWER COMMENTS

Reviewer #1 (Remarks to the Author):

In my estimation this paper is a major advance in the field of origami and should certainly be published in Nature Communications after some minor adjustments. The work is well organized, very thorough, and is of very high quality (figures and videos are outstanding). My questions and concerns are few since the paper did such a good job covering the topic. Those I have are as follows:

1) I'd be curious to learn more about the hysteresis curves of your materials. How repeatably can they achieve the same loading curves in their various configurations over different cycles? Do they dissipate much energy as they deform and through what mechanism is the energy dissipated?

2) I'd like to see more description about what specific applications the proposed materials of this paper would enable and why the specific capabilities of the new materials are necessary to enable those applications.

3) The paper is not well formatted to Nature Communications so I imagine much will need to be adapted to conform to the standard journal checklist but that's just busy work.

4) Although the paper is well written, there are numerous minor errors all throughout. Examples on the first page alone include:

a. In the sentence: "When deployed, most of them cannot provide load-bearing capacity, and those that can, they do so only..." the word "they" should be removed.

b. In the sentence: "More recently, the research trust has steered..." do you mean thrust?

c. In situ, should not have a hyphen and it should be italicized.

I'd recommend a very thorough read to make sure all the errors are removed before resubmitting.

Otherwise, fantastic work and I wish the authors the best of luck!

Reviewer #2 (Remarks to the Author):

The manuscript presents a design route for cellular materials which combines concepts of origami and kirigami and is capable of yielding multi-layered metamaterials with flat-foldable reconfigurations and load-bearing capacity (i.e., kinematic rigidity) along multiple directions including the deployment direction. Methodical analysis of the kinematics of a variety of unit cells demonstrates the possibility of tunability during the deployment path, leading to different symmetries and corresponding mechanical properties. The use of a rigidity matrix for determining the mechanisms is interesting. This work is unique and novel, since it overcomes previous conflicting design principles in foldable metamaterials. Furthermore, it presents a potentially scalable route to fabricate lightweight cellular materials. The authors have done a magnificent job in thoroughly characterizing these metamaterials; the notation is concise, the analysis is rigorous, and the text is well-written.

The manuscript, its figures, and supplementary materials are of very high quality, and thus, I recommend this manuscript for publication after addressing some minor and optional comments below.

-Regarding the scaling laws for the strength of these materials, the manuscript states that relative density was tuned by varying the dimension 'a' of the primitive feature. It is not explicitly stated, but is the assumption that 'b' remained unchanged valid? It is not immediately clear if the energetic landscape (i.e., the phase diagrams) remains unchanged if 'a' is modified. For instance, won't the path to locked states change? It would be helpful if the authors could clarify this point, explaining how the designs used to obtain these scaling laws might differ.

- Typically, variations in relative density of cellular materials are achieved by 'thickening' features. In this case, thickening the constituent sheets could provide an alternate tuning knob. Here, I suspect the stiffness per unit length 'k' of the hinges would increase throughout, and the energetic landscape would not be modified significantly. Likewise, the stiffness scaling might shift only by a constant. However, even though thorough analysis of the failure modes is beyond the scope of the manuscript, could the authors comment whether thickening of the sheets could help explain why the strength scaling diverges from the stretching or bending-dominated expectation?

-In Line #46, DoFs is used without being defined beforehand (it is defined in line 87).

- In line 57 I believe the authors meant "lose" (instead of "loose")

-In line 495, I believe the manuscript should be referring to Fig. S14, not S4.

Reviewer #3 (Remarks to the Author):

The research introduces a class of reconfigurable rigidly foldable units that can be aggregated to create lattice kirigami structures. The base unit starts with a primitive N -sided polygon and can be easily controlled to generate a variety of rigid-foldable spatial kirigami structures.

They show that the system can have lockable and flat foldable kinematic modes. Also, the increase in the number of aggregated units decreases the kinematic degrees of freedom of the system. The study explores methods to predict the lockable, flat foldable, and irregular modes of folding for multi-layered systems. The relationship between the number of stacked layers, the sides of the initial primitive polygonal pattern, and the folding modes are particularly interesting.

To the reviewer's knowledge, this particular folding pattern and its reconfigurability have been addressed previously in the literature, and it is undoubtedly worth publishing. In the introduction section, I would only recommend citing the following article, "Algorithmic lattice kirigami: A route to pluripotent materials" (doi: 10.1073/pnas.1506048112). The suggested work also deals with kirigami-based aggregation of primitive patterns in creating lattice systems.

It has been shown that increasing the number of sides of the primitive polygon N increases the number of lockable and flat-foldable modes exponentially. The work supports the conclusions and claims, and I do not suggest further evidence. The methodology is sound and meets the standards of the field. The details provided in the method section can be used to reproduce the work.

Response to the Reviewers' comments for manuscript

(Tracking #: NCOMMS-21-40077):

“Reprogrammable cellular origami: a rigidly flat-foldable class of lockable metamaterials with topological stiff states”

We thank the reviewers for their valuable feedback and constructive comments. Our detailed response (black) to each of their remarks (red) is provided below along with our revised portions of the manuscript (blue).

Response to Reviewer # 1

In my estimation this paper is a major advance in the field of origami and should certainly be published in Nature Communications after some minor adjustments. The work is well organized, very thorough, and is of very high quality (figures and videos are outstanding). My questions and concerns are few since the paper did such a good job covering the topic. Those I have are as follows:

We thank the Reviewer for his/her positive and insightful comments that enabled us to further improve the quality of our manuscript.

Comment 1.1

1) I'd be curious to learn more about the hysteresis curves of your materials. How repeatably can they achieve the same loading curves in their various configurations over different cycles? Do they dissipate much energy as they deform and through what mechanism is the energy dissipated?

Answer: Thank you for pointing out this aspect. We have performed an additional set of experiments under compressive cycles up to 75% of the specimen strength and added hysteresis curves into Fig. 5e and f in the revised manuscript. The results are for two representative cases of $\hat{N}_4n_6(7,7)$ and $\hat{N}_6n_6(7,7)$ at their densest lock configurations \mathcal{A}^2 and \mathcal{A}^3 , respectively.

Fig. 5e and f show that a large amount of energy is dissipated during the first loading-unloading cycle. After the first cycle, the energy loss diminishes significantly within the 2nd and 3rd cycles. The response almost stabilizes at the 4th cycle for $\hat{N}_4n_6(7,7)$ and $\hat{N}_6n_6(7,7)$, and thereafter no significant change can be observed in the slope of the loading/unloading curves and the amount of energy dissipation. In addition, a noticeable drop in the magnitude of the stress can be observed at the instant of the load reversal within the first 3 cycles; this drop is small after the 4th

cycle (2.5%, a value that becomes less than 1% after the 7th cycle). We attribute this phenomenon to the viscoelastic-viscoplastic behavior of our cellulose-base paperboard material [R1], an observation that parallel that found in a viscoelastic polymer tested under cyclic compressive loading [R2]. Reference [R2] is now included in the revised manuscript as reference [45].

We speculate that the underlying mechanism of the large energy loss in the first loading/unloading cycle can be mainly due to the localized plasticity and damage irreversibly occurring in the ligaments of the fold-lines (hinges). We recall that the creases are made by first perforating and then folding the paper, a process that breaks or pulls-out cellulose fibers in the stretched layers (due to a large bending stress) of the paperboard ligaments, thus making the fold-lines much weaker than the rest of the panels. When we compress our sample for the first time, the weak creases, which are prone to high stress concentrations due to perforation sites, can plastically deform. Panels can also slightly reconfigure (rotate) to accommodate the plastic deformation at the hinges. This may explain the permanent deformation (0.04 strain) we observe after the first cycle. The energy dissipation we observe in subsequent cycles might be attributed to the frictional contact between the touching panels and the viscoelastic and plastic nature of the base material (paperboard). The former may also explain why the loading and unloading modulus differ for the cycles that follow the first one.

We also remark that in the set of additional experiments that we present here, the hysteresis curves are generated at a small strain range before the stress peak, i.e., prior to panel buckling. We did so to show the repeatability of the loading curve and the energy dissipation before our lattice yields. Similarly, the hysteresis curves can be obtained at larger strain values in the post-buckling region. This aspect, the understanding of the deformation/failure mechanism, and the role of nonlinearities and defects requires an in-depth investigation, which is the subject of ongoing research and beyond the scope of this work.

To address the reviewer's comment, we have now added the following paragraph with two related sub-figures.

Fig. 5e, f show the cyclic loading-unloading curves of $\hat{N}_4 n_6(7,7)$ and $\hat{N}_6 n_6(7,7)$ under compression. Given their similarity, only the response of $\hat{N}_4 n_6(7,7)$ is examined. The hysteretic behavior might be attributed to the friction of faces and edges coming to contact, the viscoelastic-viscoplastic nature of the base-material (cellulose paperboard) as well as the local accumulation of plastic damage, which is also responsible for progressive softening. This is caused by the repeated strain of the weak crease ligaments which amplify their detrimental effect at each cycle until the response stabilizes. This occurs after approximately 4 cycles. Thereafter, no appreciable softening can be observed at higher strains nor significant variations of the modulus. At the end of the test, a strain of 0.04 is registered, indicating a permanent set not fully recovered even after several hours from the test end. These characteristics are qualitatively comparable with those observed in soft polymeric lattices exhibiting viscoelasticity and localized

plasticity under cyclic loading⁴⁵. Further work is required to quantitatively assess the response we observed and the role of the governing factors.

Comment 1.2

2) I'd like to see more description about what specific applications the proposed materials of this paper would enable and why the specific capabilities of the new materials are necessary to enable those applications.

Answer: We envision our class of foldable metamaterials to add multidirectional load bearing functionality to the existing concepts typically used for deployable structures, actuators, and robotic systems. While current origami and kirigami-based metamaterials can reconfigure, they cannot withstand any load along the direction of deployment. This limits their use as structural materials especially in an environment imposing the need to sustain volumetric pressure, either externally or internally applied. Examples of the former include remotely operated vehicles, such as shape-changing vessels and submarines, made of structural components that need to deploy underwater and provide multidirectional stiffness and strength to resist water pressure during operation. Examples of the latter include inflatable systems, such as air tents, inflatable shelters and buildings, saving boats and vests, which can not only be packed into a flat or other compact flat-folded configurations but also maintain their deployed state if punctured. Our metamaterials can thus work as the skeleton of a puncture-resistant inflatable systems that lock in place after deployment, without collapse or losing their functionality due to unforeseen deflation. In addition, their capacity to switch their permeability in situ while preserving their stiffness can be utilized for devices with porosity-modulation used in civil engineering and smart-valves for medical implants where control of the fluid flow can be done on site without the application of external occlusion elements.

To address the reviewer's comment, we have now added the following paragraph closing the discussion:

Our foldable metamaterials offer functionalities that can be used as lightweight deployable and self-locking materials, low-volume transportable packaging, actuators and lockable robotic systems that tune stiffness upon actuation. Their rigid-foldability also enables the adoption of stiff materials other than paper³⁹. In addition, the load-bearing capacity in their densest lock configuration is not limited to any specific directions, rather it is offered along their three main directions, paving the way to their use as omnidirectionally structural, yet rigidly flat-foldable mechanical metamaterials. This aspect is distinct from current origami metamaterials featuring a trade-off between reconfigurability and load-bearing capacity, the latter being limited to certain

directions^{13,24-30,32,34,38,40,42}. These concepts when used to withstand volumetric pressure in their deployed state or another set of multidirectional forces, would collapse as they remain floppy at least in one-direction. Volumetric pressure can be either externally or internally applied. Examples of the former include remotely operated vehicles, such as shape-changing vessels and submarines, made of structural components that need to deploy underwater and provide multidirectional stiffness and strength to resist water pressure during operation. Examples of the latter include inflatable systems, such as air tents, inflatable shelters and buildings, saving boats and vests, which can not only be packed into a flat or other compact flat-folded configurations, but also safely maintain their deployed state if punctured. Our metamaterials can thus work as the skeleton of a puncture-resistant inflatable systems that lock in place after deployment, without collapse or losing their functionality due to unforeseen deflation. Furthermore, the capacity to switch permeability while remaining stiff can find application in the design of adaptive porous medium and breathable walls for civil engineering, or as smart-valves for medical implants where fluid flow could be modulated *in situ* by structure rather than by external occlusions.

Comment 1.3

3) The paper is not well formatted to Nature Communications so I imagine much will need to be adapted to conform to the standard journal checklist but that's just busy work.

Answer: At the initial stage of submission, Nat. Com. does not enforce any format. We have now formatted the entire manuscript to comply to their guidelines and requirements.

Comment 1.4

4) Although the paper is well written, there are numerous minor errors all throughout. Examples on the first page alone include:

a. In the sentence: "When deployed, most of them cannot provide load-bearing capacity, and those that can, they do so only..." the word "they" should be removed.

b. In the sentence: "More recently, the research trust has steered..." do you mean thrust?

c. In situ, should not have a hyphen and it should be italicized. I'd recommend a very thorough read to make sure all the errors are removed before resubmitting.

Otherwise, fantastic work and I wish the authors the best of luck!

Answer: Thank you for pointing out these typos. We have now fixed them and carefully proofread the entire manuscript.

Response to Reviewer # 2

The manuscript presents a design route for cellular materials which combines concepts of origami and kirigami and is capable of yielding multi-layered metamaterials with flat-foldable reconfigurations and load-bearing capacity (i.e., kinematic rigidity) along multiple directions including the deployment direction. Methodical analysis of the kinematics of a variety of unit cells demonstrates the possibility of tunability during the deployment path, leading to different symmetries and corresponding mechanical properties. The use of a rigidity matrix for determining the mechanisms is interesting. This work is unique and novel, since it overcomes previous conflicting design principles in foldable metamaterials. Furthermore, it presents a potentially scalable route to fabricate lightweight cellular materials. The authors have done a magnificent job in thoroughly characterizing these metamaterials; the notation is concise, the analysis is rigorous, and the text is well-written. The manuscript, its figures, and supplementary materials are of very high quality, and thus, I recommend this manuscript for publication after addressing some minor and optional comments below.

Answer: We are pleased the Reviewer appreciates our work and thank her/him for the constructive comments.

Comment 2.1

-Regarding the scaling laws for the strength of these materials, the manuscript states that relative density was tuned by varying the dimension 'a' of the primitive feature. It is not explicitly stated, but is the assumption that 'b' remained unchanged valid? It is not immediately clear if the energetic landscape (i.e., the phase diagrams) remains unchanged if 'a' is modified. For instance, won't the path to locked states change? It would be helpful if the authors could clarify this point, explaining how the designs used to obtain these scaling laws might differ.

Answer: Thank you for pointing out the role played by the geometric parameters in the scaling laws. We first clarify that, in our work, we decided to consider a representative scenario described by patterns with geometrical parameters $a = b$ or more precisely, $\bar{b} = \frac{b}{a} = 1$. This is also stated in the last paragraph of the section “Geometry of reconfigurable unit chain”, where the notation \hat{N}_N is adopted to refer to patterns with $\bar{b} = 1$ and $\phi = \pi/3$.

For the experiments, we fabricated samples with the following geometric parameters $a = b = 10$ mm, 15 mm, and 20 mm. We have now added the information $a = b$ in the revised manuscript. Our scaling laws therefore applies to the case $\bar{b} = 1$. The deformation mechanism of our systems in the lock state depends on the nodal, edge and face connectivity. Once the material locks into its most dense configuration under compression, the face and edge connectivity are the highest. For a N_N unit, if we change \bar{b} , when $\phi = \pi/3$, the lock geometry changes as some edges would

not entirely lie on the other edges, and the edges would not meet at the same point. Therefore, their edge connectivity reduces while their face connectivity remains constant. We thus deem that their deformation and failure mechanisms might change and the scaling laws might be affected by changing the parameter \bar{b} . We speculate that the power in their stiffness vs relative density relation $E^* \propto \bar{\rho}^a$ might increase, i.e., $a > 1$. Future work is required to quantitatively address this point.

As per the phase diagrams, the plots, which are function of \bar{b} and ϕ , are also given for $\bar{b} = b/a = 1$ and $\phi = \pi/3$. Depending upon the value of \bar{b} , two general cases may arise: (I) $\bar{b} = 1$ (when $a = b$) and (II) $\bar{b} \neq 1$ (when $a \neq b$). For the former, the focus of our current study, the phase diagram does not depend on the geometric parameter a (or b); this conclusion can be reached by observing the Supplementary Equations (S42) to (S57). For the latter, the plots will scale nonlinearly with the parameter \bar{b} , yet the characteristics of the plots do not change. For instance, the biaxiality ratio that brings our materials to their lock states \mathcal{A}^2 and \mathcal{A}^3 remain constant (see, for example, Supplementary Equation (S56)), as opposed to the magnitude of the locking load. This consideration implies that the path to the lock or any other possible configurations is independent of a and b , or more generally \bar{b} . What change with the geometric parameter, ϕ , is only the slope of the lines separating regions (modes) and the biaxiality ratio r_B . Hence our methodology is not restricted to the case where $\bar{b} = b/a = 1$.

To address the reviewer's comment, we have now added the following

“geometric parameter $\bar{b} = \frac{b}{a} = 1$ ”

and

“The mode-phase diagrams depend on the geometric parameters \bar{b} and ϕ only, the former being a nonlinear scale factor, and the latter governing the slope of the lines separating mode-regions and the loading path (biaxiality ratio r_B) of a given mode (Supplementary Discussion “Energy analysis”).”

Comment 2.2

- Typically, variations in relative density of cellular materials are achieved by ‘thickening’ features. In this case, thickening the constituent sheets could provide an alternate tuning knob. Here, I suspect the stiffness per unit length ‘k’ of the hinges would increase throughout, and the energetic landscape would not be modified significantly. Likewise, the stiffness scaling might shift only by a constant. However, even though thorough analysis of the failure modes is beyond the scope of the manuscript, could the authors comment whether thickening of the sheets could help explain why the strength scaling is diverges from the stretching or bending-dominated expectation?

Answer: We agree with the reviewer's comment that the thickening of the panels is a route to increase the relative density, hence slightly enhancing the stiffness per unit length ' k ' of hinges. However, our hinges are generated by perforating and then folding the paperboard, thus becoming much weaker than the base material especially upon loading due to damage and plasticity. For this reason, we deem scaling the hinge size will provide a very mild increase in the hinge stiffness, thereby generating not appreciable divergence from the scaling laws for stiffness. On the front of failure, however, our hinges can still play a role, which is not clearly quantified by our experiments and measurements. We speculate that the failure mechanism and strength are the results of the combined effect of panel failure by either yield or buckling, hinge failure, and defects both in the base material and the architecture. The competition between these factors is also influenced by the actions of the adjacent panels. In the lock configuration, each parallelogram panel is confined along their short diagonal by the edge of two adjacent parallelogram panels (see for example Fig. 1a in the main manuscript). From the experimental response, we observe the failure strength appears with the local buckling of the panels. We conclude the faces are not fully confined (against buckling), hence the yield failure of the stretching dominated lattices ($\sigma_Y \propto \bar{\rho}$) cannot be observed. Neither, the faces are completely free to buckle ($\sigma_Y \propto \bar{\rho}^3$) due to the confining action exerted by the adjacent faces. In contrast, we observe a strength scaling law ($\sigma_Y \propto \bar{\rho}^2$) with exponent between buckling and yield of panels. Further work is required to fully understand and formulate the deformation and failure mechanism we observed during our experiments.

Comment 2.3

-In Line #46, DoFs is used without being defined beforehand (it is defined in line 87).

Answer: Agreed. We have now defined *DoFs* when it appears for the first time in the text.

Comment 2.4

- In line 57 I believe the authors meant "lose" (instead of "loose")

Answer: Agreed. The typo is fixed in the revised manuscript.

Comment 2.5

-In line 495, I believe the manuscript should be referring to Fig. S14, not S4.

Answer: Corrected. Thank you.

Response to Reviewer # 3

The research introduces a class of reconfigurable rigidly foldable units that can be aggregated to create lattice kirigami structures. The base unit starts with a primitive N -sided polygon and can be easily controlled to generate a variety of rigid-foldable spatial kirigami structures. They show that the system can have lockable and flat foldable kinematic modes. Also, the increase in the number of aggregated units decreases the kinematic degrees of freedom of the system. The study explores methods to predict the lockable, flat foldable, and irregular modes of folding for multi-layered systems. The relationship between the number of stacked layers, the sides of the initial primitive polygonal pattern, and the folding modes are particularly interesting. To the reviewer's knowledge, this particular folding pattern and its reconfigurability have been addressed previously in the literature, and it is undoubtedly worth publishing.

Answer: We appreciate the Referee's positive feedback and recommendation.

Comment 3.1

In the introduction section, I would only recommend citing the following article, "Algorithmic lattice kirigami: A route to pluripotent materials" (doi: 10.1073/pnas.1506048112). The suggested work also deals with kirigami-based aggregation of primitive patterns in creating lattice systems.

Answer: Thank you for the suggestion. We agree with the reviewer and have now added that reference in the introduction section of the revised manuscript.

Comment 3.2

It has been shown that increasing the number of sides of the primitive polygon N increases the number of lockable and flat-foldable modes exponentially. The work supports the conclusions and claims, and I do not suggest further evidence. The methodology is sound and meets the standards of the field. The details provided in the method section can be used to reproduce the work.

Answer: We thank the Referee for the positive feedback.

[R1] Tjahjanto, D. D., Girlanda, O., & Östlund, S. (2015). Anisotropic viscoelastic–viscoplastic continuum model for high-density cellulose-based materials. *Journal of the Mechanics and Physics of Solids*, 84, 1-20.

[R2] Gavazzoni, M., Foletti, S., & Pasini, D. (2022). Cyclic response of 3D printed metamaterials with soft cellular architecture: The interplay between as-built defects, material and geometric non-linearity. *Journal of the Mechanics and Physics of Solids*, 158, 104688.

REVIEWERS' COMMENTS

Reviewer #1 (Remarks to the Author):

Given the changes made to the manuscript I now believe the manuscript is ready for publication. I congratulate the authors.

Reviewer #2 (Remarks to the Author):

The revised manuscript satisfactorily addresses all point set forth in the review. The added experiments and main-text elaborations have strengthened the manuscript. The current version of the manuscript is appropriate for publication in Nature Communications. Congratulations to the authors on a great contribution to the literature.